# Evaluation of Semantic Segmentation Methods for Land Use with Spectral Imaging Using Sentinel-2 and PNOA Imagery

Oscar D. Pedrayes [1,*], Darío G. Lema [1], Daniel F. García [1], Rubén Usamentiaga [1] and Ángela Alonso [2]

1 Department of Computer Science and Engineering, Campus de Viesques, University of Oviedo, 33204 Gijón, Spain; UO243567@uniovi.es (D.G.L.); dfgarcia@uniovi.es (D.F.G.); rusamentiaga@uniovi.es (R.U.)

2 Department of Spatial Data, Seresco S.A., Matemático Pedrayes 23, 33005 Oviedo, Spain; angela.alonso@seresco.es

* Correspondence: UO251056@uniovi.es

**Abstract:** Land use classification using aerial imagery can be complex. Characteristics such as ground sampling distance, resolution, number of bands and the information these bands convey are the keys to its accuracy. Random Forest is the most widely used approach but better and more modern alternatives do exist. In this paper, state-of-the-art methods are evaluated, consisting of semantic segmentation networks such as UNet and DeepLabV3+. In addition, two datasets based on aircraft and satellite imagery are generated as a new state of the art to test land use classification. These datasets, called UOPNOA and UOS2, are publicly available. In this work, the performance of these networks and the two datasets generated are evaluated. This paper demonstrates that ground sampling distance is the most important factor in obtaining good semantic segmentation results, but a suitable number of bands can be as important. This proves that both aircraft and satellite imagery can produce good results, although for different reasons. Finally, cost performance for an inference prototype is evaluated, comparing various Microsoft Azure architectures. The evaluation concludes that using a GPU is unnecessarily costly for deployment. A GPU need only be used for training.

**Keywords:** sentinel; pnoa; sigpac; unet; deeplab; multi-spectral; aerial; agriculture; convolutional neural network; semantic segmentation

## 1. Introduction

Currently, when there is a need for land use classification from aerial imagery, the standard procedure consists in a manual process carried out by professionals in that specific field. This process is time consuming and costly. Automation both reduces costs and makes the process much faster. New applications that require a response time of milliseconds can be developed, for example, for the classification of each frame of a video, and thus they are able to process constantly changing regions.

Being able to automate location tasks in aerial imagery, such as land use or crop classification, opens up the possibility of exploring new services, such as crop monitoring, which may be of interest to companies. Many satellites not only update their data every few days but also allow free access. This could lead to significant advances in agricultural applications and other fields.

In other works [1–5], whenever a study of a land use classification dataset is performed, the datasets usually remain private, so the results proposed are not reproducible. There does not appear to be a common dataset specifically for testing accuracy and performance of land use classification, in the way COCO [6] and PASCAL VOC [7] are used for image classification. For this reason, two datasets are offered in this work for public use. Both use data from SIGPAC [8], a free Spanish government database for agricultural land identification, to generate the ground truths required for training and testing the models. The first dataset, called UOS2, obtains its source imagery from the Sentinel-2 satellite [9].

The second dataset, called UOPNOA, obtains its source imagery from the aircraft imagery from the National Plan for Aerial Orthophotography [10] , also known as PNOA, a free governmental database of aerial orthophotography of Spain. These datasets are released with this work for public use and can be downloaded with the following DOI (last accessed on 10 June 2021): doi.org/10.5281/zenodo.4648002.

Both UOPNOA and UOS2 identify eleven land use types, which means eleven different target classes. They are constructed from SIGPAC data of the same region, the northern part of the Iberian Peninsula plateau in Spain. The two datasets are very different as one of them has been obtained from aircraft imagery and the other from satellite imagery. However, as both of them are generated from the same region, a realistic comparison can be made. Obviously, the aircraft images are taken much closer to the ground, which means that objects and textures have a much better resolution. In contrast, the dataset taken from satellite imagery has more bands, offering up to thirteen different bands.

Automation of land use classification using satellite imagery is not a new concept [3,11–13]. The previous methods used are mainly based on Random Forest [14] and Support Vector Machine [4,15]. There are over 18,000 articles in 2020 mentioning Random Forest and land use classification and 11,000 articles mentioning SVM and land use classification in 2020. However, there are better alternatives with simple convolutional neural networks [1,16]. These networks are viable as long as the dataset used to train the models has sufficient data and variability, which may be an issue in areas where the data is more restricted. Complex convolutional neural networks using specifically semantic segmentation networks perform even better if a correct dataset is provided, as has been demonstrated with other satellite imagery tasks such as land cover classification [5].

This has led to the study of convolutional neural networks and the development of semantic segmentation for aerial imagery [17]. There are scientific publications comparing these methods with more recent semantic segmentation architectures in satellite imagery [5,18]. However, there are few publications that study the latest advances in semantic segmentation in the specific application of land use classification tasks and their particular behavior using this type of dataset. Those that do exist identify only a few main target classes as abstract as "vegetation" [19] or highly differentiated classes such as "Bare Rock", "Beaches", "Water bodies", etc. For this reason, UNet [20] and DeepLabV3+ [21], two semantic segmentation architectures, are tested with the UOPNOA and UOS2 datasets. In this way, the influence of the different characteristics of the images in semantic segmentation networks is studied.

The aim of this paper is to evaluate the difference between satellite imagery and aircraft imagery for the specific task of land use classification using semantic segmentation methods, the generation of a dataset of each type from the same region, and the evaluation of a service for a realistic use case. Parameters such as ground sampling distance, multispectral bands, and complexity of target classes are taken into account.

The accuracy of land use classification depends on the ground sampling distance of the images and the number of bands. The UOS2 uses images with up to thirteen bands, including different infrared bands. These images have bands with a ground sampling distance or GSD of 10, 20, or 60 m/pixel, meaning that there are 10, 20, or 60 m of ground between the center of one pixel and the center of the next. UOPNOA only includes RGB bands, but its GSD is of 0.25 m/pixel. To isolate the effects of the number of bands and to determine the minimum number needed to ensure good results, several versions of the UOS2 dataset are used: two versions of three bands (RGB and non-RGB), six bands, ten bands and thirteen bands. Since UOPNOA has a much smaller GSD than UOS2, results from the two datasets are compared to determine its influence. The greater the GSD, the more difficult it is to differentiate one class from another, even to the human eye. Finally, as having too many classes can be detrimental to the accuracy of the models, tests are conducted with different numbers of classes. Similar classes are merged to see how far two similar classes can be separated without confusion in classification.

This study is performed with semantic segmentation networks, testing well-known [22–26] convolutional neural networks such as UNet [20] or DeepLabV3+ [21]. This means that not only are different imagery sources tested but also different methods. In addition, the complexity of the target classes is studied. Different numbers of classes, generic and specific classes, and the use of an all-purpose class are also evaluated in this study.

After training and testing, the feasibility of deployment is evaluated for the most relevant implementations. A microservices architecture following modern standards is then deployed to create an inference prototype on different infrastructures. Thus, the performance and cost of different deployments can be studied. The deployments compared are local, infrastructure as a service (IaaS), container as a service (CaaS), and function as a service (FaaS). The hardware and its components are also compared, including the use of a GPU versus a CPU, to determine whether the added cost of a GPU is justified.

In summary, the contributions offered by this work are the following:

- Different sources of imagery such as aircraft and satellite imagery are compared.
- Two new public datasets to evaluate land use classification with different characteristics and complexities are released for public use.
- Differences in GSD, number of bands of the images, resolution, and other characteristics are evaluated for semantic segmentation for land use classification.
- Semantic segmentation methods are evaluated in aerial and satellite imagery.
- The complexity of classes and target classes is studied.
- Cost performance of different deployments for a possible implementation of a semantic segmentation model for land use classification in aerial imagery is analyzed.

## 2. Materials and Methods

### 2.1. Datasets

#### 2.1.1. Target Classes

"Sistema de Información Geográfica de Parcelas Agrícolas" or SIGPAC, is a free database provided by the Spanish government which allows one to geographically identify the plots declared by the farmers. It has up to thirty different classes defined, but according to the requisites of the potential users of this kind of product, not all of these classes are relevant for agriculture. From SIGPAC the following classes are extracted, each of them associated to a color. Both UOPNOA and UOS2 datasets will use the same classes and color palette.

- UN—Unproductive
- PA—Pastureland
- SH—Shrub grass
- FO—Forest
- BU—Buildings and urban area
- AR—Arable Land
- GR—Grass with trees
- RO—Roads
- WA—Water
- FR—Fruits and nuts
- VI—Vineyard

Pixels that do not correspond with any of these classes will either be assigned to an all-purpose class called "OT—Other" or be unused depending on the experiment. This class is of no relevance to the study and its only purpose it to provide a realistic prediction that permits pixels that do not belong to any other target class.

To test a lower number of more generic classes, new classes made from combinations of the previous one are created. This allows for simpler experiments to study class complexity.

- PASHGR—All pastures
- BURO—All infraestructures

- ▪ ARVI—Arable lands and vineyards

### 2.1.2. Uopnoa Dataset

PNOA is a database of digital aerial orthophotographs that are accessible for free in Enhanced Compressed Wavelet, a format capable of compressing enormous images and storing their georeference. Georeference is important to establish the ground truth for a dataset as it makes it possible to merge data from different sources such as SIGPAC. This format is necessary because of the resolution of the images, with hundreds of thousands of pixels in width and height, but it can be converted to GEOTIFF if required. Each orthophotograph has a GSD of 0.25 m/pixel and covers a region equivalent to one MTN50 page, the National Topographic Map of Spain. One of the main advantages of PNOA, apart from its great GSD, is that the images have no clouds or other defects. However, it has a great disadvantage for periodic crop monitoring as this data is only updated once a year. Information about the bands is described in Table 1.

**Table 1.** Bands from PNOA.

| Bands | GSD (m/pixel) | Bits |
| --- | --- | --- |
| B1 Red | 0.25 | 8 |
| B2 Green | 0.25 | 8 |
| B3 Blue | 0.25 | 8 |
| B4 NIR | 0.25 | 8 |

UOPNOA consists of 33,699 images of 256 × 256 pixels. These images are cropped out of PNOA images that cover a region equivalent to an MTN50 page. In order to keep the georeferencing data in the images, functions from the GDAL library are used to crop the images.

Images from PNOA are downloaded using CNIG (Centro Nacional de Información Geográfica) [27], as this is the official procedure of the government of Spain to download PNOA imagery. However, the band "B4 NIR" cannot be downloaded.

The annotation process was carried out using the coordinates of the SIGPAC plots to colour the regions of the images to generate masks. Since the images are georeferenced, this process can be done automatically.

Figure 1 shows one of these cropped images. To check that the ground truth has been correctly generated, it is compared to official data from the SIGPAC visor. A screenshot of this visor is presented next to the ground truth mask. The SIGPAC visor separates each plot, or individually fenced piece of land, but as only the type of land use is needed, there is no need to separate plots with the same type.

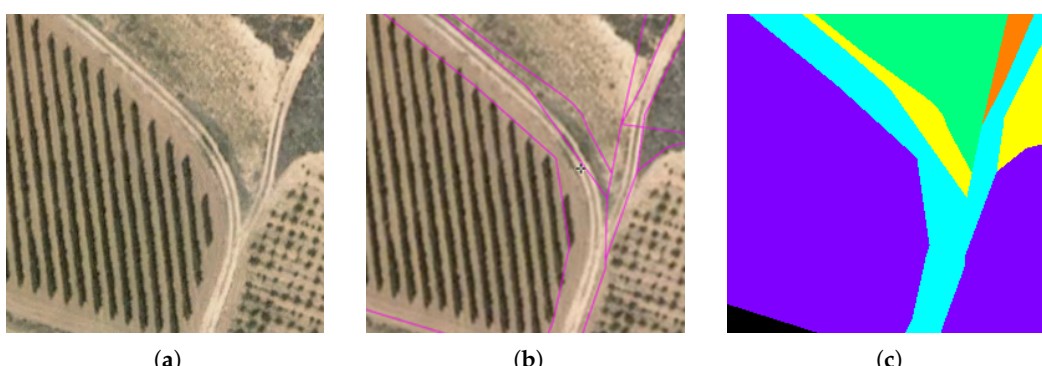

(**a**)　　　　　　　　　　(**b**)　　　　　　　　　　(**c**)

**Figure 1.** Cropped image and ground truth checking for UOPNOA. (**a**) cropped image of 256 × 256 pixels from a PNOA image, (**b**) SIGPAC visor of the same region to check the ground truth mask , (**c**) ground truth mask.

The number of plots from SIGPAC used to make the ground truth for each class is presented in Table 2 along with the number of pixels of the UOPNOA dataset. A visual representation of the pixels is shown in Figure 2. A large difference between the AR class and the rest of the classes can be observed. This kind of land use is usually bigger than the rest of the target classes, and in the region selected this class is very common.

**Table 2.** Number of SIGPAC plots and pixels for each class used in UOPNOA.

| Class | Plots | # of Pixels |
|-------|-------|-------------|
| UN | 410 | 8,095,351 |
| PA | 1883 | 46,738,269 |
| SH | 3467 | 168,342,415 |
| FO | 472 | 73,980,816 |
| BU | 125 | 7,562,696 |
| AR | 4935 | 968,106,602 |
| GR | 220 | 93,208,409 |
| RO | 943 | 67,552,872 |
| WA | 340 | 19,776,364 |
| FR | 93 | 4,150,529 |
| VI | 1759 | 163,774,218 |

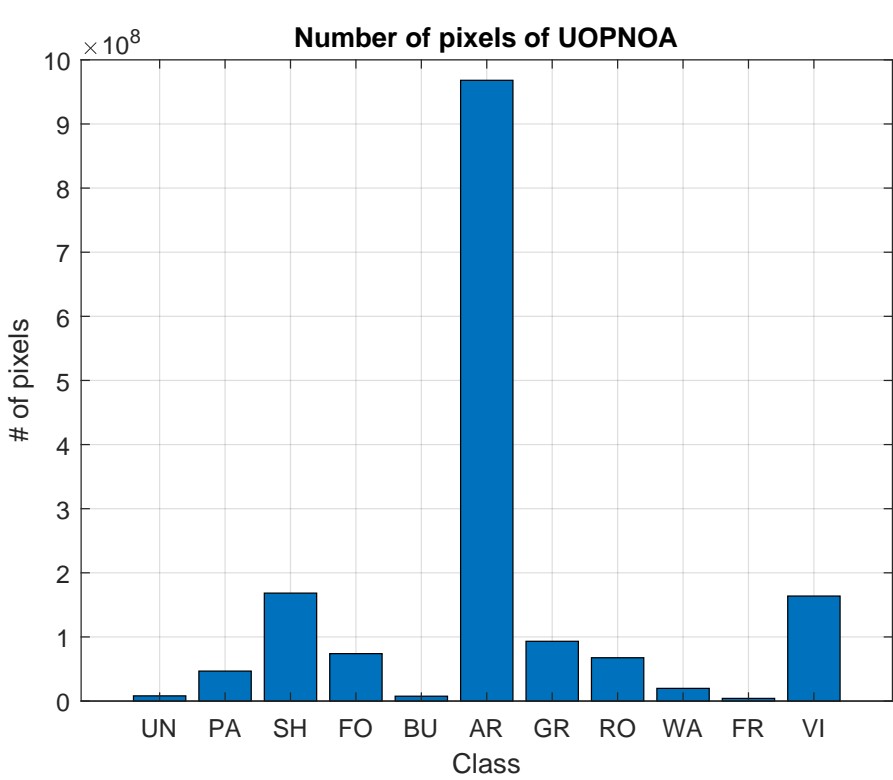

**Figure 2.** Number of pixels for each class of the dataset UOPNOA.

### 2.1.3. Uos2 Dataset

Sentinel-2 is a mission based on a constellation of two identical satellites (Sentinel-2A and Sentinel-2B) on the same orbit, which offers its data for free in GEOTIFF. This format is capable of storing not only the image with as many bands as necessary but also all the data needed to georeference every pixel in the image. These georeferences are used to establish the ground truth from SIGPAC data. Sentinel-2 offers thirteen different bands, with different wavelengths, bandwidths, and GSDs. Each satellite has a 10-day period around the equator and 4–6 days at midlatitudes. However, since they are half a revolution apart, the time required to update the images is reduced by half.

Sentinel-2 has been widely used in works that study semantic segmentation of satellite imagery, as there are more than 11,000 papers mentioning this satellite in 2020 and more than 38,000 in total. One of its main advantages is its period of five days around the equator and 2–3 days at midlatitudes.

In Table 3, the data for each band is shown. The columns "Wavelength" and "Bandwidth" have information from Sentinel-2A/Sentinel-2B in this same order.

**Table 3.** Bands from Sentinel-2A and Sentinel-2B.

| Band | Wavelength (nm) | Bandwidth (nm) | GSD (m/pixel) |
|---|---|---|---|
| B1 Coastal aerosol | 442.7/442.2 | 21/21 | 60 |
| B2 Blue | 492.4/492.1 | 66/66 | 10 |
| B3 Green | 559.8/559.0 | 36/36 | 10 |
| B4 Red | 664.6/664.9 | 31/31 | 10 |
| B5 VRE | 704.1/703.8 | 15/16 | 20 |
| B6 VRE | 740.5/739.1 | 15/15 | 20 |
| B7 VRE | 782.8/779.7 | 20/20 | 20 |
| B8 NIR | 832.8/832.9 | 106/106 | 10 |
| B8A Narrow NIR | 864.7/864.0 | 21/22 | 20 |
| B9 Water vapour | 945.1/943.2 | 20/21 | 60 |
| B10 SWIR Cirrus | 1374.0/1376.9 | 31/30 | 60 |
| B11 WIR | 1614.0/1610.4 | 91/94 | 20 |
| B12 SWIR | 2202.0/2185.7 | 175/185 | 20 |

UOS2 consists of 1958 images of 256 × 256 pixels, all of them taken in July 2020. These images are cropped from Sentinel-2 images. To facilitate the generation of ground truth masks, images from Sentinel-2 that cover entire regions from MTN50 pages and its respective data from SIGPAC are obtained. Combining these to make the ground truth is straightforward. Then, these images are cropped to images of 256 × 256 pixels.

Images from Sentinel-2 are downloaded using SentinelHub [28], as this is the easiest way to download a specified region from a concrete date using a simple script.

To obtain valid images from Sentinel-2, images that do not contain clouds or any other defect are searched for manually. When a region of interest includes anything that could compromise its quality, another date on which the image has no defects is found. In this manner, the number of images to check goes from 1958 small images to 39 larger Sentinel-2 images, making it much easier to check manually.

The annotation process was carried out similarly to the UOPNOA dataset, using the coordinates of the SIGPAC plots to paint the mask accordingly.

Figure 3 shows an example of one these cropped images and its mask. In the same way as before, to check that the ground truth is correctly generated, it is compared with the official data from the SIGPAC visor.

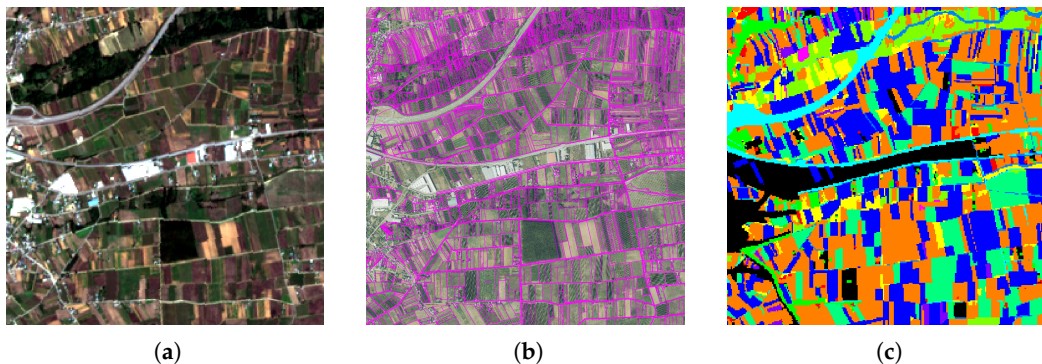

(**a**)　　　　　　　　　　　　　(**b**)　　　　　　　　　　　　　(**c**)

**Figure 3.** Cropped image and ground truth checking for UOS2. (**a**) cropped image of 256 × 256 pixels from a Sentinel-2 image, (**b**) SIGPAC visor of the same region , (**c**) ground truth mask.

The number of plots from SIGPAC used to make the ground truth for each class is presented in Table 4 along with the number of pixels of the UOS2 dataset. A visual representation of the pixels is in Figure 4. Given that the UOS2 dataset covers the same region as the UOPNOA dataset, a large difference between the AR class and the rest of the classes is also observed.

**Table 4.** Number of SIGPAC plots used in UOPNOA.

| Class | Plots | # of Pixels |
|-------|-------|-------------|
| UN | 26,912 | 1,455,995 |
| PA | 152,359 | 5,591,140 |
| SH | 265,046 | 16,465,578 |
| FO | 48,977 | 10,746,146 |
| BU | 207,839 | 1,047,900 |
| AR | 772,427 | 60,046,203 |
| GR | 28,649 | 4,494,801 |
| RO | 73,250 | 3,719,742 |
| WA | 16,802 | 1,733,641 |
| FR | 6306 | 566,355 |
| VI | 87,071 | 4,126,803 |

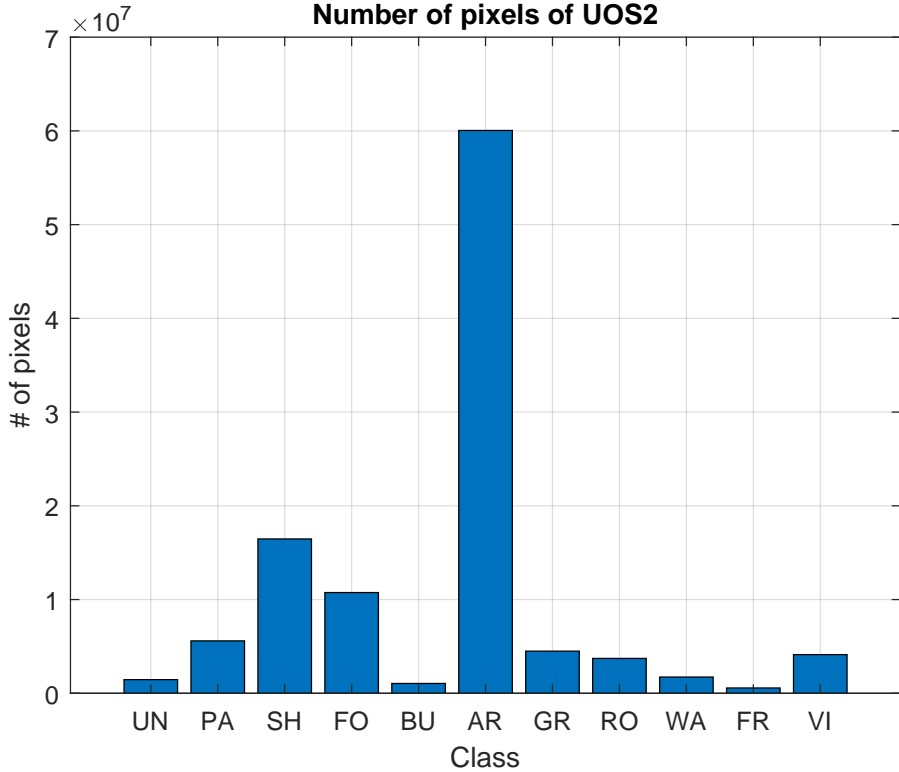

**Figure 4.** Number of pixels for each class of the dataset UOS2.

## 2.2. Analysis of the Evaluated Architectures

### 2.2.1. Unet

Originally developed for binary classification to segment cells in biomedical images, UNet is one of the most widely referenced networks in semantic segmentation, cited in over 25,000 papers. Its original motivation was to train and produce precise predictions with as few training images as possible. The name UNet comes from its symmetric encoder-decoder architecture, giving it a u-shaped architecture. As it has a relatively simple architecture, it offers a high degree of flexibility. Thanks to this flexibility, many networks based on its architecture have been developed. UNet was quickly modified to work with all kinds of images and numbers of classes and is currently the most widely used with satellite imagery.

Therefore, UNet will be used in this evaluation study. In Figure 5, an overview of UNet architecture can be seen.

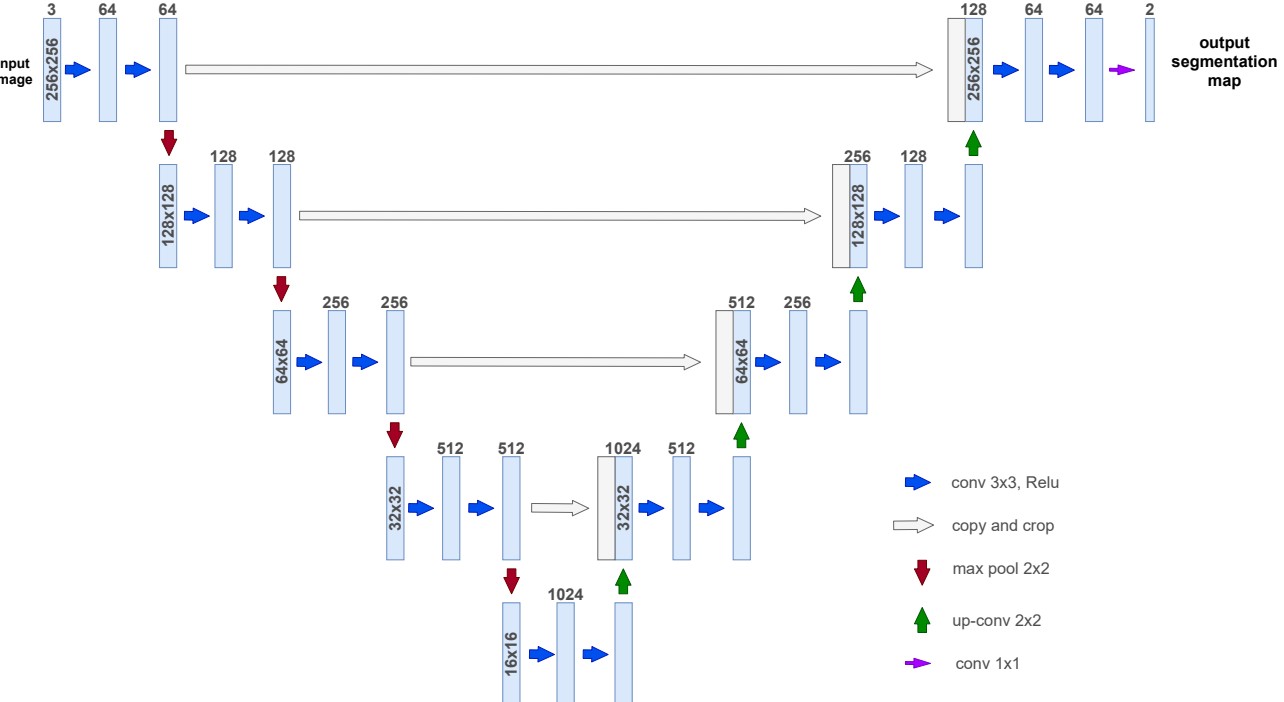

**Figure 5.** UNet architecture. This diagram is based on the original UNet publication [20].

### 2.2.2. Deeplab

Developed by Google, this network is still under development, having had a number of versions until now. The first version [29] uses atrous convolutions to control the resolution at which feature responses are computed. The second version, DeepLabV2 [30], adds an Atrous Spatial Pyramid Pooling (also known as ASPP) module to make better predictions at different scales. The third version, DeepLabV3 [31], upgrades the ASPP module and also adds Batch Normalization to the architecture to make setting up the training easier, since a manual normalization is no longer needed. The fourth version, DeepLabV3+ [21], adds a decoder module to convert its architecture to encoder-decoder. Finally, there is also an auto machine learning version called AutoDeepLab [32], whose architecture is based on DeepLabV3+. AutoDeepLab is not evaluated in this work because its computational cost would make the time required to train it far too long. Therefore, because DeepLabV3+ is the most recent version and usually obtains better results than UNet, it has been selected for evaluation in this paper. This architecture is widely used since it has over 3600 citations on its paper. In Figure 6, an overview of DeepLabV3+ architecture can be seen.

### 2.3. Network and Training Parameters

In this section the following parameters used in the experiments to train the models and to modify the architecture of the networks are described:

- Input size: The resolution and number of channels of the input images.
- Classes: Number of classes to train.
- Backbone network: Classification network used as a part of the initial architecture of a more complex network, such as DeepLabV3+.
- Depth: The number of max pooling layers in the UNet architecture.
- Filters on first level: The number of filters on the first convolution of the UNet architecture. This value is multiplied by 2 on every depth level.

- Output stride: The division between the input image resolution and the final feature map. For example, if the input image has a resolution of $256 \times 256$ and the final feature map $32 \times 32$, then the output stride is 8. It controls the separation between each step of the convolution. This is a configurable parameter in the Google DeepLabV3+ implementation.
- Padding: A filler that is added to each convolution so as not to reduce the resolution of the final feature map.
- Class balancing: The method used to prevent a biased training when the dataset is unbalanced.
- Solver: Algorithm that calculates the gradient when training the network.
- Epochs: Number of times the complete dataset is used in training.
- Fine-Tune Batch Normalization: A parameter that allows the DeepLabV3+ implementation to train the batch normalization layers instead of using the pretrained ones.
- BatchSize: Number of images used in each batch. Since the entire dataset cannot be stored in memory, the dataset is divided into batches.
- LearningRate: Parameter that controls how the network weights are adjusted with respect to the gradient.
- Gradient clipping: Limits the maximum value of the gradient to prevent the exploding gradient problem when training.
- L2 regularization: A technique to reduce the complexity of a model by penalizing the loss function. As a result, overfitting is reduced.
- Data augmentation: Generation of new data from the original data.
- Shuffle: The dataset is shuffled on every epoch.

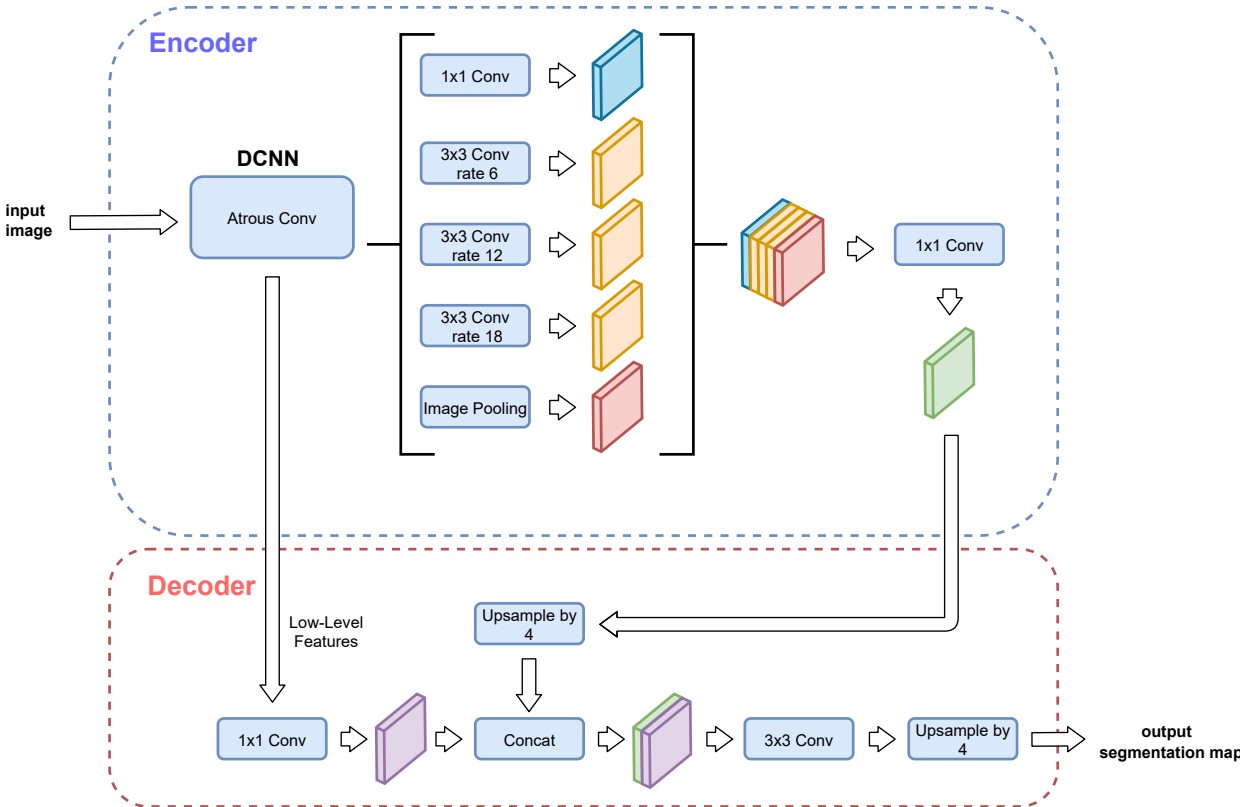

**Figure 6.** DeepLabV3+ architecture. A backbone network such as ResNet-101 or Xception65 can be used as DCNN. This diagram is based on the original DeepLabV3+ publication [21].

### 2.4. Performance Metrics

To evaluate accuracy of the trained models, a brief description of the metrics used [33] is presented below. These metrics are usually calculated per class and then averaged to obtain a global metric for the model. This is done to prevent unbalanced classes from affecting the global results.

- True positive (*TP*): number of pixels that are correctly classified.
- True negative (*TN*): number of pixels that are from other classes and are correctly classified as such.
- False positive (*FP*): number of pixels classified as the target class but belonging to other classes.
- False negative (*FN*): number of pixels that are classified as other classes but are from the target class.
- Producer accuracy (*PA*): Percentage of correctly predicted pixels of a given class. Producer accuracy is often called Recall (*R*).

$$PA = \frac{TP}{TP + FN} \tag{1}$$

- User accuracy (*UA*): A percentage that represents how many predictions are correct from the total number of predictions for that class. User accuracy is often called Precision (*P*).

$$UA = \frac{TP}{TP + FP} \tag{2}$$

- F-score (*F1$_1$*): equivalent to the Dice Coefficient, is a metric that combines both Producer accuracy (Recall) and User accuracy (Precision) as a way to represent them as a single value, making comparisons between models easier.

$$F_1 = \frac{2 \times UA \times PA}{UA + PA} \tag{3}$$

- Overall accuracy (*OA*): A percentage that represents how many pixels are correctly classified from the total. Overall accuracy is often called Global accuracy (*GA*). This metric can be misleading if the classes are not balanced. For example, given two classes, if one of them represents 99% of the pixels in the dataset and the other one represents the remaining 1%, even if all of the pixels from the second class are classified wrongly as pixels from the first class, this metric will still obtain an *OA* of 99%.

$$OA = \frac{TP}{Total\ of\ pixels} \tag{4}$$

- Intersection-Over-Union (*IoU*): equivalent to the Jaccard Index, this metric measures the degree of similarity between the ground truth and prediction sets.

$$IoU = \frac{Area\ of\ Overlap}{Area\ of\ Union} = \frac{TP}{TP + FN + FP} \tag{5}$$

### 2.5. Experimental Procedure

In order to make a comparison between methods, a basic experimentation with RF and SVM has been performed. This experimentation consists in using the values Red, Green, and Blue as features. For the RF and SVM experiments, a total of 12,000 pixels were used, 1000 pixels for each class. The entire dataset has not been used due to temporal constraints and because these models do not scale to the same level as a neural networks, so no improvement is observed above a certain amount of data.

The hyperparameters of all networks must be tuned to match the dataset used. In this work these hyperparameters are tuned manually in each network for the two datasets

separately. In the case of UNet, these hyperparameters are: learning rate, epochs, L2 regularization, depth levels, number of filters in the convolutional layers, type of solver, gradient clipping, and batch size. On the other hand, for DeepLabV3+, the hyperparameters used are: learning rate, epochs, L2 regularization, type of solver, usage of fine-tune batch norm, gradient clipping, batch size, and the backbone network used.

Every parameter is tested one by one until its best configuration is found. The effect of changing multiple parameters at the same time has not been tested. In this regard there is still leeway for further improvement. As this process is done manually, this methodology is the best option to obtain an optimal configuration.

There must be sufficient separation between the training and testing data to avoid repeating samples. To obtain more meaningful and realistic results, the testing must be done with data that the model has not seen before. To do this separation, the dataset is divided into four parts. In this way, not only can a separation between training and testing data be made but also a four-fold crosstesting.

For the selection of the optimal hyperparameters, instead of evaluating each class separately only the global User and Producer accuracy will be compared. For the best experiments, both Producer accuracy and User accuracy for each class will be considered and studied.

The visualization examples from some of the testing data give a better idea of how well the network performs. These images allow for a more in depth analysis and might give explanations for some of the results obtained. To distinguish when the "Other" class is being used, pixels identified with this class are coloured black. Similarly, when no class is associated with a pixel, these pixels are coloured white. Pixels not classified as belonging to any class are not taken into account when training or testing the network. This means that for the purposes of the evaluation, these pixels and their predictions are irrelevant.

The machines used for the experiments consist of a GPU NVIDIA RTX 2080 Ti and a I7-9700K CPU.

## 3. Results And Discussion

### 3.1. Previous Methods

To make a fair comparison and see the performance of the proposed methods, they should be compared to the previous methods: Random Forest and Support Vector Machine. Both methods need much less data than a common neural network and take much longer to run. For this research work a reduced UOPNOA dataset with 1000 samples (pixels) per class was used, with a total of 12,000 samples. More detailed results can be found in Table 5. Random Forest achieved results of 0.074% overall accuracy with only four seconds of training. The Support Vector Machine achieved 0.073% overall accuracy in three and a half minutes of training. The time required to train these experiments does not scale linearly: when tested with only 6000 samples, 500 per class, SVM took only about thirty seconds. The overall accuracy did not improve, obtaining almost the same value of 0.07%. The testing of the two methods coincides with the same set of images that the rest of the UOPNOA experiments use.

**Table 5.** Global metrics for the experiments with RF and SVM on UOPNOA.

| Experiment | *OA* | *PA* | *UA* | *IoU* | $F_1$ |
|:---:|:---:|:---:|:---:|:---:|:---:|
| RF | 0.074 | 0.114 | 0.130 | 0.034 | 0.121 |
| SVM | 0.073 | 0.138 | 0.130 | 0.035 | 0.134 |

The features used for these experiments consist of the red, green, and blue values of each pixel. Proper experimentation with these methods must include feature engineering. This is not necessary with neural networks, as they generate their own features. This is a great advantage, although a large dataset is needed. When comparing Random Forest and

SVM with proper feature engineering with neural networks, results can still be worse as long as the dataset used is sufficiently large and variable [22,34,35].

*3.2. Experimentation with Uopnoa*

Experiments with DeepLabV3+ and UNet for the UOPNOA dataset and a discussion of their results are presented in this subsection.

3.2.1. Deeplabv3+

Optimal network configuration for DeepLabV3+ with the dataset UOPNOA is presented in Table 6, and its best training parameters are shown in Table 7. The architecture of the network used for this experiment is the official Google architecture. It can be downloaded from its repository on GitHub along with pretrained models to reduce training time. After manually tuning the network to work with this dataset, the optimal hyperparameters found are listed in Table 6.

**Table 6.** Network parameters for DeepLabV3+ on UOPNOA.

| Network Parameters | |
| --- | --- |
| Input size | $256 \times 256 \times 3$ |
| Classes | 11 |
| Backbone network | Xception41 |
| Output stride | 16 |
| Padding | Yes |
| Class balancing | Median frequency weighting |

The input size from the evaluation for UNet was maintained for all the experiments in this work to compare the exact same dataset between implementations. The following backbone networks were tested: Xception65, Xception41, Xception71, MobileNetV2, MobileNetV3 Small, MobileNetV3 Large, and Resnet50. For each of them, the best learning rate, batch size, and number of epochs was established. The values tested for output stride are 8 and 16, the most commonly used. Padding is always added to equal the sizes of the outputs to the inputs, making tasks such as translating information to geojson easier. For class balancing, "No class weighting", "inverse frequency weighting", and "median frequency weighting" are compared.

**Table 7.** Training parameters for DeepLabV3+ on UOPNOA.

| Training Parameters | |
| --- | --- |
| Solver | Adam |
| Epochs | 60 |
| Fine Tune Batch Normalization | No |
| Batch size | 12 |
| Learning rate | 0.00005 |
| Gradient clipping | No |
| L2 regularization | 0.0004 |
| Shuffle | Yes |
| Data augmenting | No |

For the training parameters in Table 7, the network solvers Adam [36] and SGDM [37] were both evaluated. The number of epochs for training was calculated by overshooting their values and then observing where the overfitting starts or where the loss seems to plateau. The use of Fine-Tune Batch Normalization was tested with multiple configurations to ensure that it does not depend on parameters such as the backbone network, learning rate, batch size, or epochs. Batch size is the parameter with the greatest effect. With small batch sizes, the use of Fine-Tune Batch Normalization makes the results noticeably better, but with bigger batch sizes, in the order of twelve images, results are slightly worse.

DeepLabV3+ is a very complex architecture but a batch of twelve images can be used for all the backbone networks evaluated for eleven gigabytes of VRAM. The more images used in a single batch, the better the results. Batch sizes in the range of four to sixty-four were tested for those backbone networks that allow it. Constant learning rates from 0.001 to 0.00005 were studied using decrements that consist of halving or dividing by ten its value for each experiment, depending on its loss during training. The training process did not encounter the exploding gradient problem so gradient clipping was not needed.

Two experiments were carried out to compare the use of the class "Other". The first experiment, called "Base", had no "Other" class, whereas the second experiment, called "All-Purpose", did. Both experiments used the same data and configurations.

Global metrics for both experiments can be seen in Table 8. In the "Base" experiment, a considerable increase in PA and UA was observed, with approximately 10% and 8% better results for PA and UA respectively. Moreover, the OA improves by up to 15% when the "Other" class is not used. This indicates that an all-purpose class does not help in this kind of dataset and network.

**Table 8.** Global metrics for the experiment with DeepLabV3+ on UOPNOA.

| Experiment | *OA* | *PA* | *UA* | *IoU* | $F_1$ |
|:---:|:---:|:---:|:---:|:---:|:---:|
| Base | 0.898 | 0.781 | 0.758 | 0.637 | 0.769 |
| All-Purpose | 0.750 | 0.678 | 0.678 | 0.524 | 0.678 |

To prove that the tuning was done correctly, progress from the loss function for training and validation can be seen in Figure 7. Training loss starts to plateau and validation loss stabilizes.

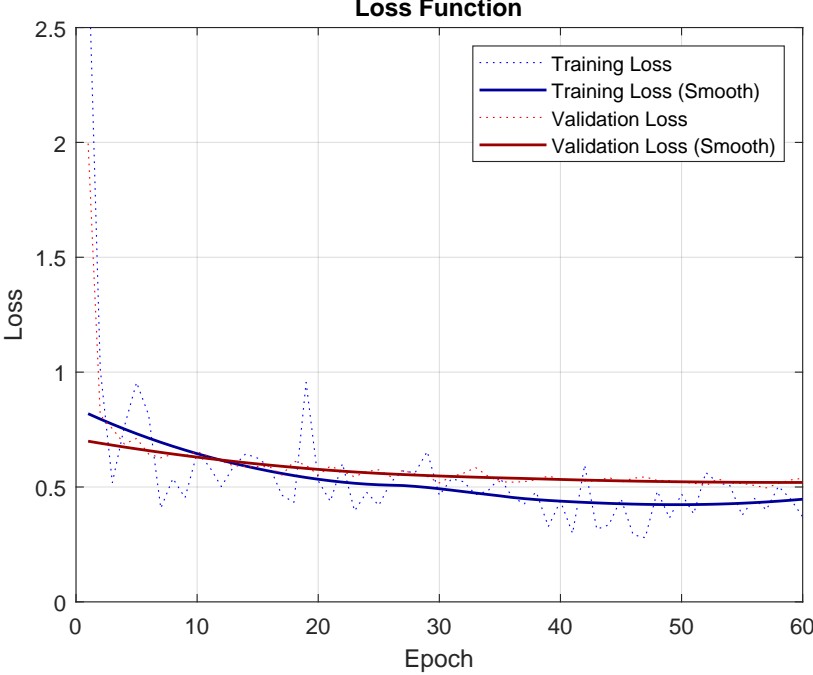

**Figure 7.** Loss function for the "Base" experiment with DeepLabV3+ for PNOA.

Looking at the results of both experiments by class in Table 9, it is obvious that in most cases an all-purpose class only introduces confusion. This may be because an all-purpose class like "Other" is too generic. This class obtains the best results of all of the classes, but since it is of no interest, this is not the desired behavior.

**Table 9.** Class metrics for the experiment with DeepLabV3+ on UOPNOA.

| Class | Experiment Base | | | Experiment All-Purpose | | |
|-------|-----|-----|-----|-----|-----|-----|
| | *PA* | *UA* | *IoU* | *PA* | *UA* | *IoU* |
| UN | 0.56 | 0.66 | 0.43 | 0.65 | 0.63 | 0.47 |
| PA | 0.79 | 0.78 | 0.65 | 0.55 | 0.48 | 0.27 |
| SH | 0.58 | 0.53 | 0.38 | 0.38 | 0.39 | 0.30 |
| FO | 0.84 | 0.84 | 0.73 | 0.63 | 0.68 | 0.49 |
| BU | 0.85 | 0.86 | 0.76 | 0.78 | 0.75 | 0.62 |
| AR | 0.94 | 0.97 | 0.92 | 0.76 | 0.82 | 0.65 |
| GR | 0.75 | 0.89 | 0.69 | 0.81 | 0.87 | 0.72 |
| RO | 0.84 | 0.60 | 0.54 | 0.82 | 0.75 | 0.65 |
| WA | 0.77 | 0.47 | 0.41 | 0.75 | 0.52 | 0.44 |
| FR | 0.66 | 0.73 | 0.53 | 0.46 | 0.56 | 0.34 |
| VI | 0.96 | 0.95 | 0.92 | 0.62 | 0.77 | 0.53 |
| OT | - | - | - | 0.86 | 0.86 | 0.76 |

From Figures 8 and 9 the two experiments can be compared graphically. At first glance, both seem to make good predictions, especially when all the pixels of an image are of the same class. However, upon closer inspection the "Base" experiment is sightly more stable and closer to a human technician's classifications. This confirms the figures presented in Table 9.

### 3.2.2. Unet

The optimal network configuration for UNet with the dataset UOPNOA is presented in Table 10, along with its best training parameters in Table 11. This architecture is exactly the same as the official publication [20], except that it has been adapted to take RGB images instead of grayscale images. As this problem is a multi-class classification rather than a binary classification, categorical cross entropy (CCE) is used to calculate loss values. Different numbers of depth levels, from one to five, and filters on the first level, from 16 to 128, were tested. The optimal configurations coincide with the original architecture.

**Table 10.** Network parameters for UNet on UOPNOA.

| Network Parameters | |
|-------|-------|
| Input size | $256 \times 256 \times 3$ |
| Classes | 11 |
| Depth | 4 |
| Filters on first level | 64 |
| Padding | Yes |
| Class balancing | Median frequency weighting |

After testing multiple resolutions from $1024 \times 1024$ pixels to $256 \times 256$ pixels for the input size of the images, it was found that the network performs far better when using a resolution of $256 \times 256$ pixels. To maintain cohesion and facilitate comparisons, this resolution is used throughout this work. As for class balancing, "median frequency weighting" always obtains the best results.

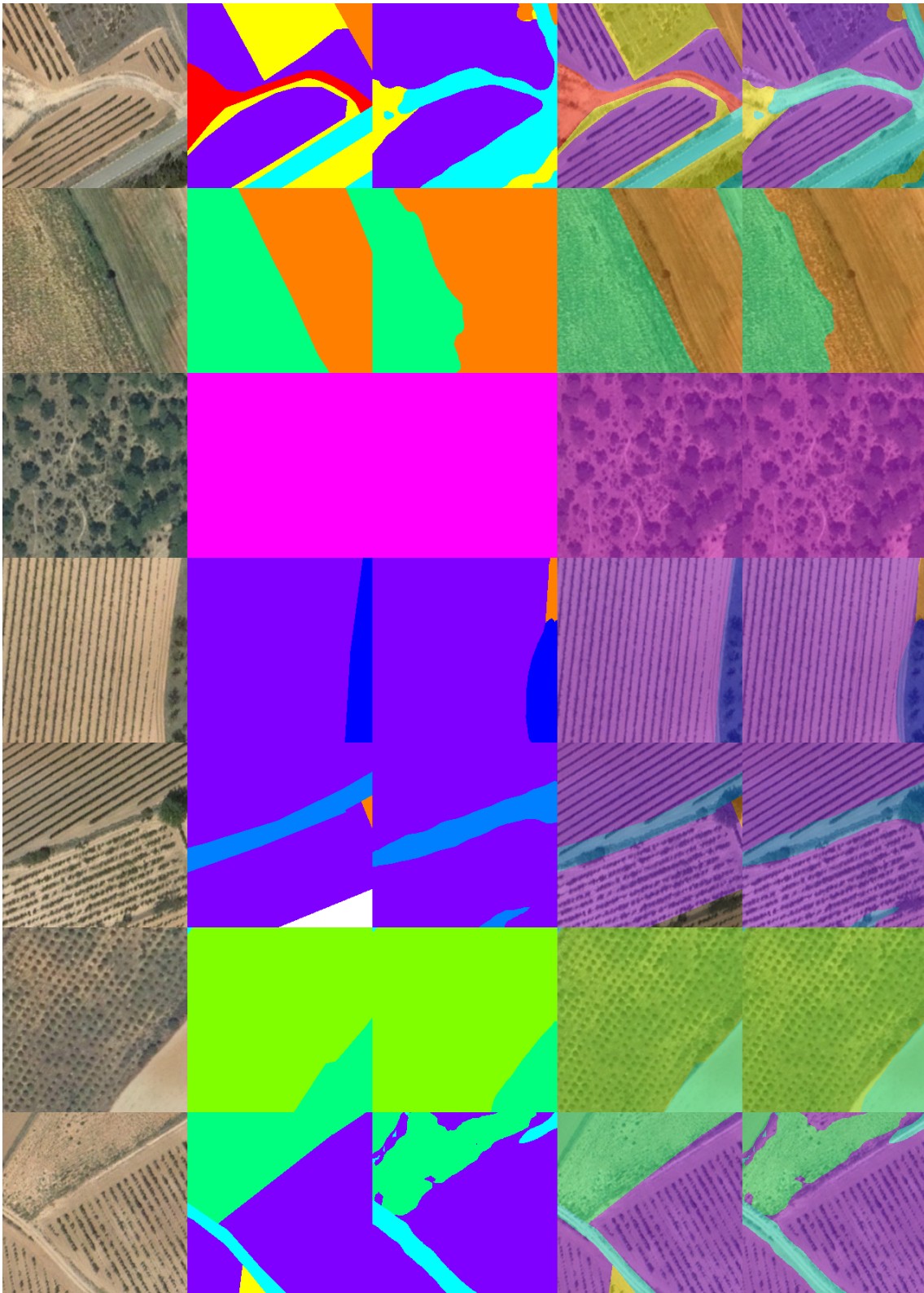

**Figure 8.** Visualization of the predicted results for DeepLabV3+ evaluated in UOPNOA (Experiment Base). (**1st col.**) Original images, (**2nd col.**) ground truth masks, (**3rd col.**) predictions, (**4th col.**) overlapping of original images with ground truth masks, (**5th col.**) overlapping of original images with predictions.



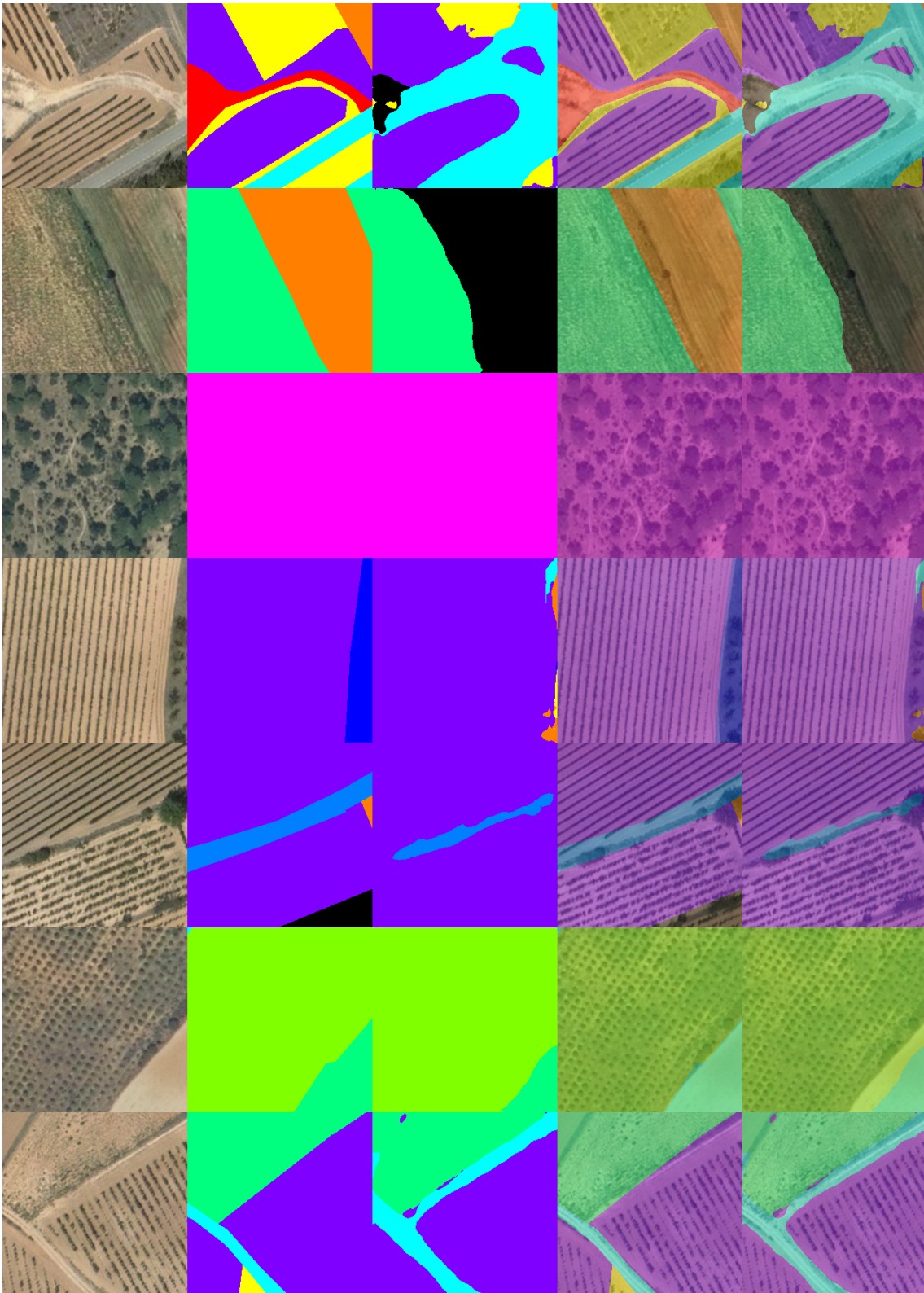

**Figure 9.** Visualization of the predicted results for DeepLabV3+ evaluated in UOPNOA with "Other" class (Experiment All-Purpose). (**1st col.**) Original images, (**2nd col.**) ground truth masks, (**3rd col.**) predictions, (**4th col.**) overlapping of original images with ground truth masks, (**5th col.**) overlapping of original images with predictions.

**Table 11.** Training parameters for UNet on UOPNOA.

| Training Parameters | |
|---|---|
| Solver | Adam |
| Epochs | 20 |
| Batch size | 16 |
| Learning rate | 0.0001 |
| Gradient clipping | 1.0 |
| L2 regularization | 0.0001 |
| Shuffle | Yes |
| Data augmenting | No |

Tuning the training parameters from Table 7 is a process equivalent to that described for the DeepLabV3+ network, except there is no backbone network or Fine-Tune Batch Normalization. Gradient clipping is not relevant as its value in this case is too high, so it does not affect training. Furthermore, it is not needed since there is no exploding gradient problem. Finally, L2 regularization was tested and a value of 0.0001 works best, reducing overfitting and improving results. Higher values were tested but caused accuracy to drop. Similarly, when using lower values, overfitting starts earlier, reducing accuracy in testing.

To compare UNet with DeepLabV3+, both experiments done with UOPNOA are recreated with the optimal configuration for UNet. This also allows for a study on the behavior of the all-purpose class "Other" in different networks.

Global metrics for both experiments can be seen in Table 12. In the experiment "Base", an increase in PA and UA is observed, around 1% and 8% better results respectively. The OA improves up to 19% when the "Other" class is not used. Like the experiments with DeepLabV3+, this indicates that an all-purpose class does not help in this kind of dataset and network. When comparing these results to DeepLabV3+, a difference of 16% and 30% in PA and UA is observed for the best experiments.

**Table 12.** Global metrics for the experiment with UNet on UOPNOA.

| Experiment | *OA* | *PA* | *UA* | *IoU* | $F_1$ |
|---|---|---|---|---|---|
| Base | 0.830 | 0.618 | 0.473 | 0.473 | 0.536 |
| All-Purpose | 0.641 | 0.607 | 0.391 | 0.391 | 0.476 |

To prove that the tuning has been done correctly, progress from the loss function for training and validation can be seen in Figure 10. Validation loss stabilizes but training loss keeps lowering; if more epochs are executed, overfitting of the model would start to occur.

In Table 13, the results from each class are presented. Like the experiments conducted on DeepLabV3+, there is a considerable increase in accuracy when the all-purpose class is not used. On the other hand, in this case the class "Other" has the worst results. Like the DeepLabV3+ experiments, the rest of the classes drop noticeably when *PA* and *UA* are considered.

Figures 11 and 12 show the predicted results from the experiments. These predictions, when compared to those of DeepLabV3+, are quite noisy, and there is far more confusion between classes. In this case the experiment "Base" is similar to the experiment "All-Purpose" for these particular examples, even though there is a difference of 8% in *UA*.

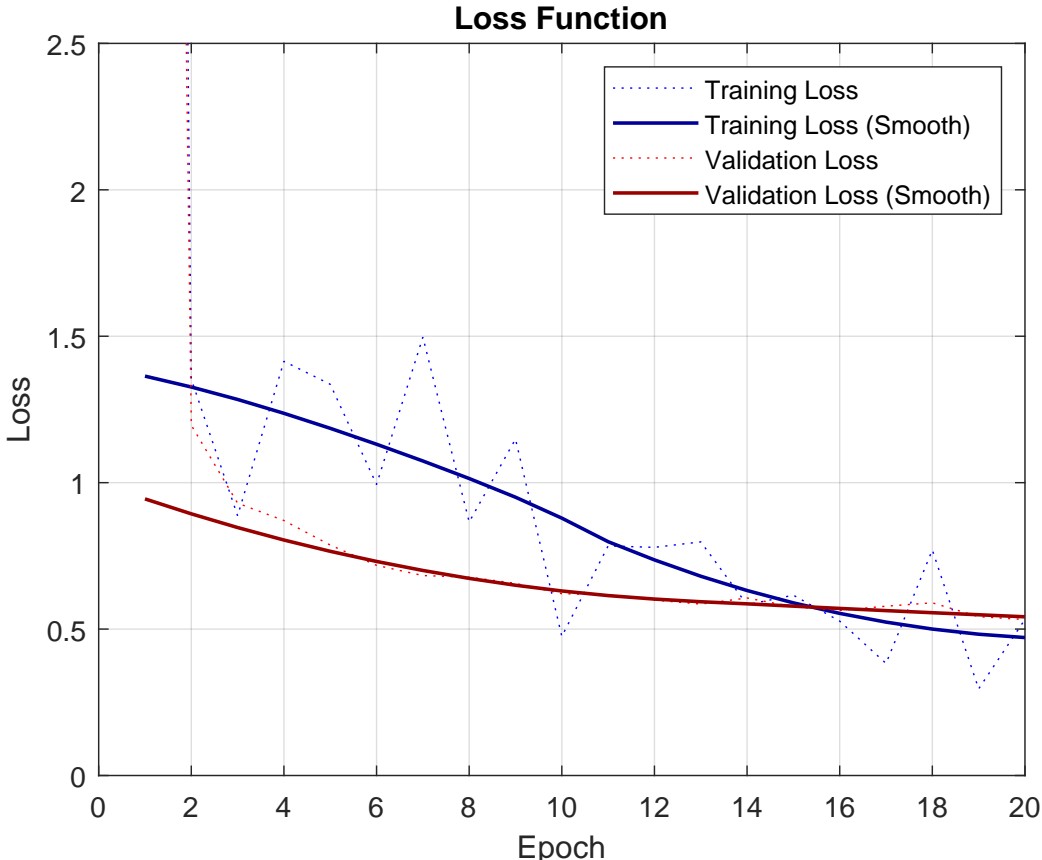

**Figure 10.** Loss function for the "Base" experiment with UNet for PNOA.

**Table 13.** Class metrics for the experiment with UNet on UOPNOA.

| Class | Experiment Base | | | Experiment All-Purpose | | |
|---|---|---|---|---|---|---|
| | *PA* | *UA* | *IoU* | *PA* | *UA* | *IoU* |
| UN | 0.45 | 0.29 | 0.21 | 0.40 | 0.23 | 0.17 |
| PA | 0.54 | 0.25 | 0.20 | 0.36 | 0.21 | 0.15 |
| SH | 0.69 | 0.62 | 0.49 | 0.64 | 0.51 | 0.39 |
| FO | 0.21 | 0.78 | 0.20 | 0.52 | 0.63 | 0.40 |
| BU | 0.59 | 0.92 | 0.56 | 0.71 | 0.62 | 0.49 |
| AR | 0.93 | 0.96 | 0.90 | 0.88 | 0.75 | 0.68 |
| GR | 0.62 | 0.70 | 0.49 | 0.67 | 0.59 | 0.46 |
| RO | 0.77 | 0.51 | 0.45 | 0.79 | 0.37 | 0.34 |
| WA | 0.60 | 0.43 | 0.34 | 0.63 | 0.30 | 0.26 |
| FR | 0.45 | 0.92 | 0.43 | 0.54 | 0.70 | 0.44 |
| VI | 0.90 | 0.97 | 0.88 | 0.95 | 0.76 | 0.74 |
| OT | - | - | - | 0.15 | 0.47 | 0.13 |

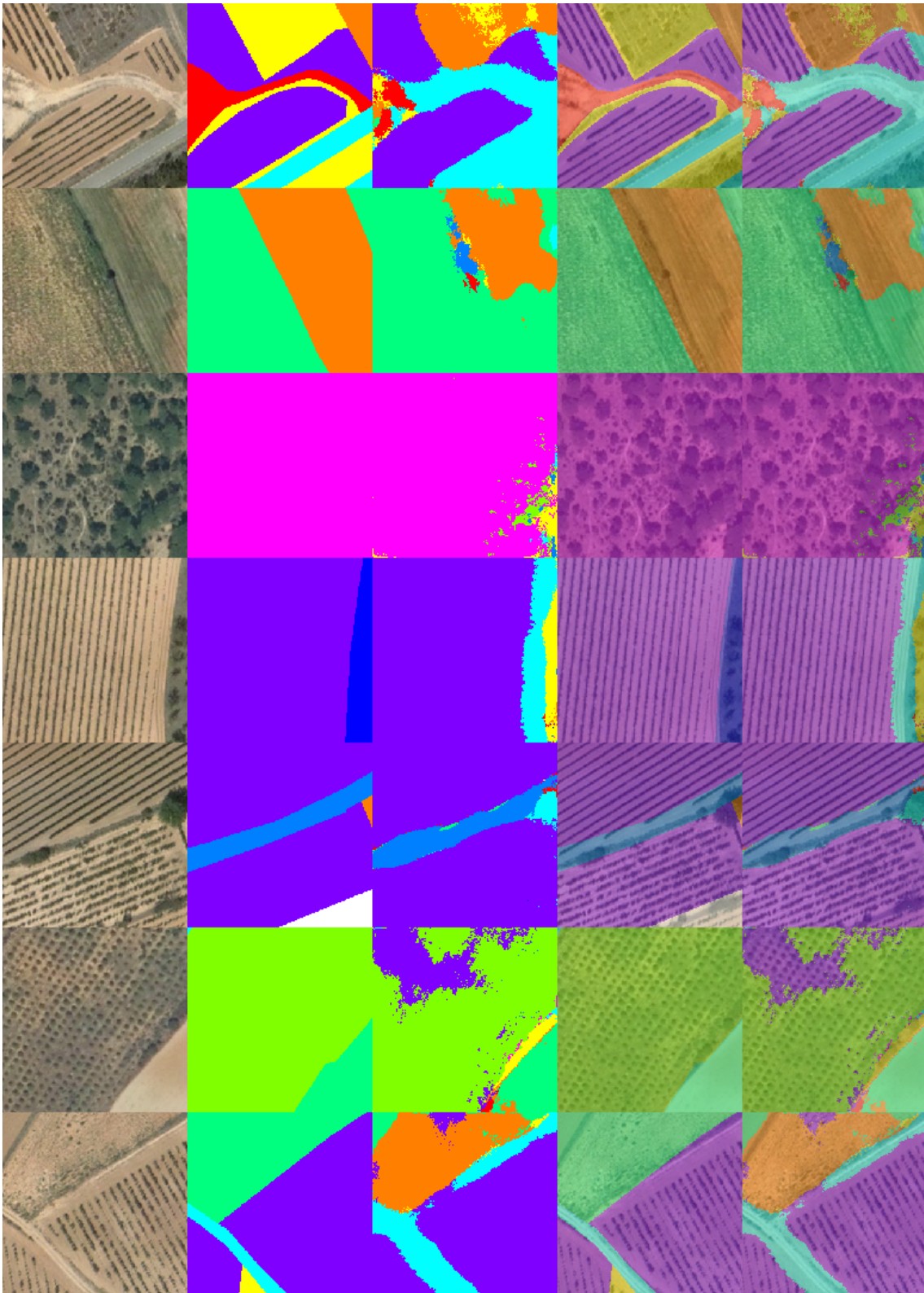

**Figure 11.** Visualization of the predicted results for UNet evaluated in UOPNOA (Experiment Base). (**1st col.**) Original images, (**2nd col.**) ground truth masks, (**3rd col.**) predictions, (**4th col.**) overlapping of original images with ground truth masks, (**5th col.**) overlapping of original images with predictions.

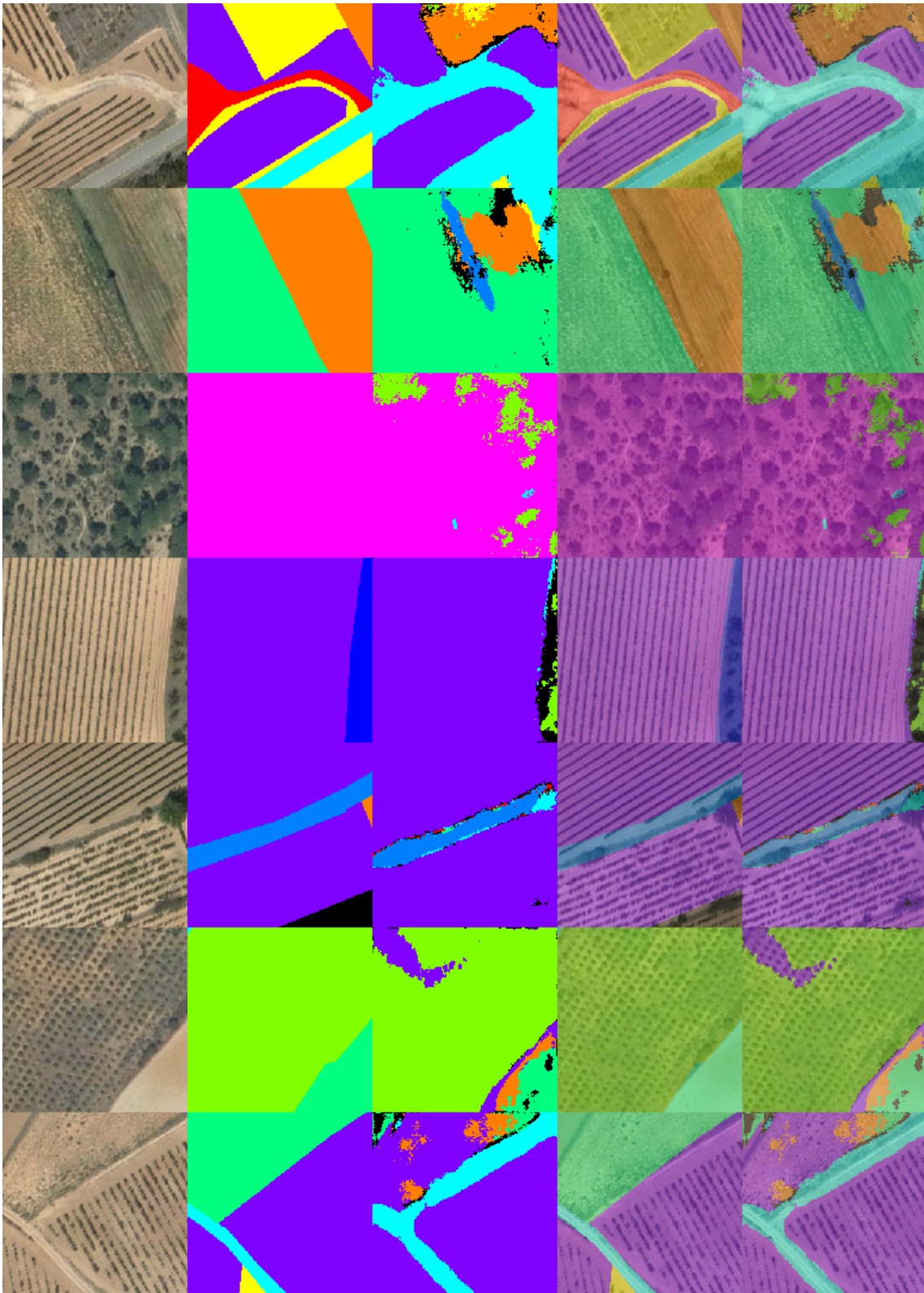

**Figure 12.** Visualization of the predicted results for UNet evaluated in UOPNOA with "Other" class (Experiment All-Purpose). (**1st col.**) Original images, (**2nd col.**) ground truth masks, (**3rd col.**) predictions, (**4th col.**) overlapping of original images with ground truth masks, (**5th col.**) overlapping of original images with predictions.

### 3.3. Experimentation with Uos2

Experiments with DeepLabV3+ and UNet for the UOS2 dataset and discussion of their results are presented in this subsection.

#### 3.3.1. Unet with All Classes

Optimal network configuration for UNet with the dataset UOS2 is presented in Table 14 and its best training parameters in Table 15. This architecture is similar to its first official publication [20], but as the images used have ten different bands, the architecture has been changed to allow these input images. In addition, the problem to solve is not a binary classification but rather a multi-class classification. The final difference from the original implementation is that rather than using sixty-four filters on the first depth level, thirty-two are used, as this is the optimal configuration found. This means that on every depth level there are half as many filters as the original implementation. As a result, far less computational resources are needed to train the network, with no tradeoff in accuracy for this particular dataset. The number of depth levels necessary were also tested, and four were found to be optimal, coinciding with the official implementation.

**Table 14.** Network parameters for UNet on UOS2.

| Network Parameters | |
|---|---|
| Input size | $256 \times 256 \times 10$ |
| Classes | 11 |
| Depth | 4 |
| Filters on first level | 32 |
| Padding | Yes |
| Class weighting | Median frequency |

**Table 15.** Training parameters for UNet on UOS2.

| Training Parameters | |
|---|---|
| Solver | Adam |
| Epochs | 125 |
| Batch size | 32 |
| Learning rate | 0.0005 |
| Gradient clipping | 1.0 |
| L2 regularization | 0.0001 |
| Shuffle | Yes |
| Data augmenting | Yes (Mirroring on both axes) |

Training parameters for Table 15 were found following the same procedure described on UNet for UOPNOA. In this case data was augmented for the training data since there were far fewer images than on UOPNOA.

After training the model, the test data was evaluated. The results obtained can be seen in Table 16. In addition to the experiments "Base" and "All-Purpose", four more experiments were carried out. These included: experiment "RGB" to test how the dataset performs with only RGB bands, experiment "ME", which utilizes only three multi-spectral bands (B8 NIR, B12 SWIR, B6 VRE) to compare with the RGB experiment, and two additional experiments using six and thirteen bands. Given that the "Base" experiment and the "All-Purpose" experiments use ten bands, a comparison between three, six, ten, and thirteen bands can be made. All the experiments except the "Base" experiment use the class "Other".

Using only three bands, both the "RGB" and "ME" experiments perform badly, obtaining a PA and UA close to 10%. Moreover, there is little difference between using RGB bands or the three multi-spectral bands selected.

The experiments with six, ten, and thirteen bands have similar results. This proves that only six bands are really needed and that three bands does not provide enough data to differentiate between classes at this GSD.

A comparison between the "Base" experiment from UNet on UOPNOA (0.61 *PA* and 0.47 *UA*) and the best experiment from UNet on UOS2 (0.56 *PA* and 0.42 *UA*) reveals that UOPNOA has better results. Thus, GSD is the most important feature of an aerial imagery dataset. The next most important factor is to have more bands than simply RGB.

Finally, the "Base" experiment outperforms the "All-Purpose" experiment in the same way that occurs in the UOPNOA dataset. This can be seen in detail in Table 17

**Table 16.** Global metrics for the experiment with UNet on UOS2.

| Experiment | *OA* | *PA* | *UA* | *IoU* | $F_1$ |
|---|---|---|---|---|---|
| Base | 0.647 | 0.569 | 0.428 | 0.323 | 0.489 |
| All-Purpose | 0.527 | 0.521 | 0.364 | 0.259 | 0.429 |
| RGB | 0.585 | 0.090 | 0.079 | 0.053 | 0.084 |
| ME | 0.570 | 0.091 | 0.109 | 0.052 | 0.099 |
| 6 bands | 0.569 | 0.480 | 0.373 | 0.252 | 0.420 |
| 13 bands | 0.589 | 0.463 | 0.371 | 0.261 | 0.412 |

**Table 17.** Class metrics for the experiment with UNet on UOS2.

| | Experiment Base | | | Experiment All-Purpose | | |
|---|---|---|---|---|---|---|
| Class | *PA* | *UA* | *IoU* | *PA* | *UA* | *IoU* |
| UN | 0.56 | 0.22 | 0.19 | 0.51 | 0.22 | 0.18 |
| PA | 0.42 | 0.28 | 0.20 | 0.46 | 0.21 | 0.17 |
| SH | 0.36 | 0.67 | 0.30 | 0.37 | 0.54 | 0.28 |
| FO | 0.59 | 0.60 | 0.43 | 0.52 | 0.48 | 0.34 |
| BU | 0.81 | 0.50 | 0.45 | 0.74 | 0.39 | 0.34 |
| AR | 0.77 | 0.94 | 0.73 | 0.69 | 0.85 | 0.62 |
| GR | 0.62 | 0.37 | 0.30 | 0.58 | 0.28 | 0.23 |
| RO | 0.36 | 0.16 | 0.12 | 0.40 | 0.13 | 0.11 |
| WA | 0.61 | 0.22 | 0.19 | 0.63 | 0.17 | 0.16 |
| FR | 0.34 | 0.10 | 0.08 | 0.45 | 0.08 | 0.07 |
| VI | 0.78 | 0.59 | 0.51 | 0.76 | 0.56 | 0.48 |
| OT | - | - | - | 0.09 | 0.38 | 0.08 |

To prove that the tuning has been done correctly, progress from the loss function for training and validation can be seen in Figure 13. Training loss and validation loss both start to stabilize.

Observing the visualization of the predictions from both experiments (Figures 14 and 15), it is clear that UOS2 is far more complex than UOPNOA. There are many different classes in a single image and their area is considerably smaller. Taking this into consideration, the results provided are good. When examining the predictions globally, they are very similar to the ground truth.

Finally, when comparing the use of the class "Other", the predictions are similar although there is a significant difference in results.

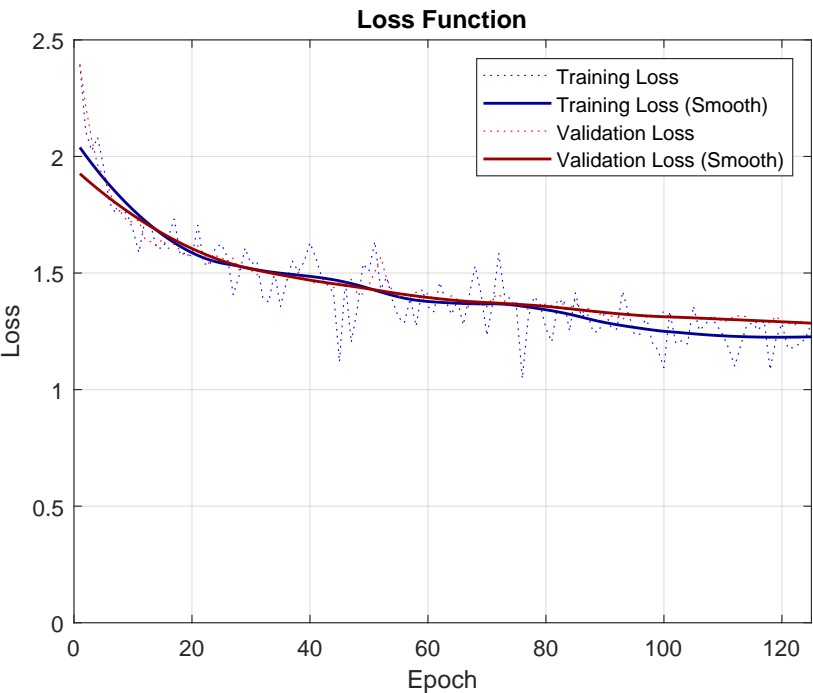

**Figure 13.** Loss function for the "Base" experiment with UNet for UOS2.

### 3.3.2. Unet with Simplified Classes

The purpose of these experiments is to test the effects of using fewer classes by merging similar classes together. In addition, the effects of including the all-purpose class "Other" with fewer and more stable classes is studied. To make these new classes, the confusion matrix from the experiments is used to distinguish which classes are the most similar. Only classes that are similar from a logical standpoint are merged. Classes that do not obtain good results and have little relevance are not used.

Optimal configurations from the previous experiments with eleven classes are reused (Tables 18 and 19). Experiments were conducted to verify that these configurations are still optimal.

**Table 18.** Network parameters for UNet with simplified classes on UOS2.

| Network Parameters | |
| --- | --- |
| Input size | $256 \times 256 \times 10$ |
| Classes | 4 |
| Depth | 4 |
| Filters on first level | 32 |
| Padding | Yes |
| Class weighting | Median frequency |

**Table 19.** Training parameters for UNet with simplified classes on UOS2.

| Training Parameters | |
| --- | --- |
| Solver | Adam |
| Epochs | 125 |
| Batch size | 32 |
| Learning rate | 0.0005 |
| Gradient clipping | No |
| L2 regularization | No |
| Shuffle | Yes |

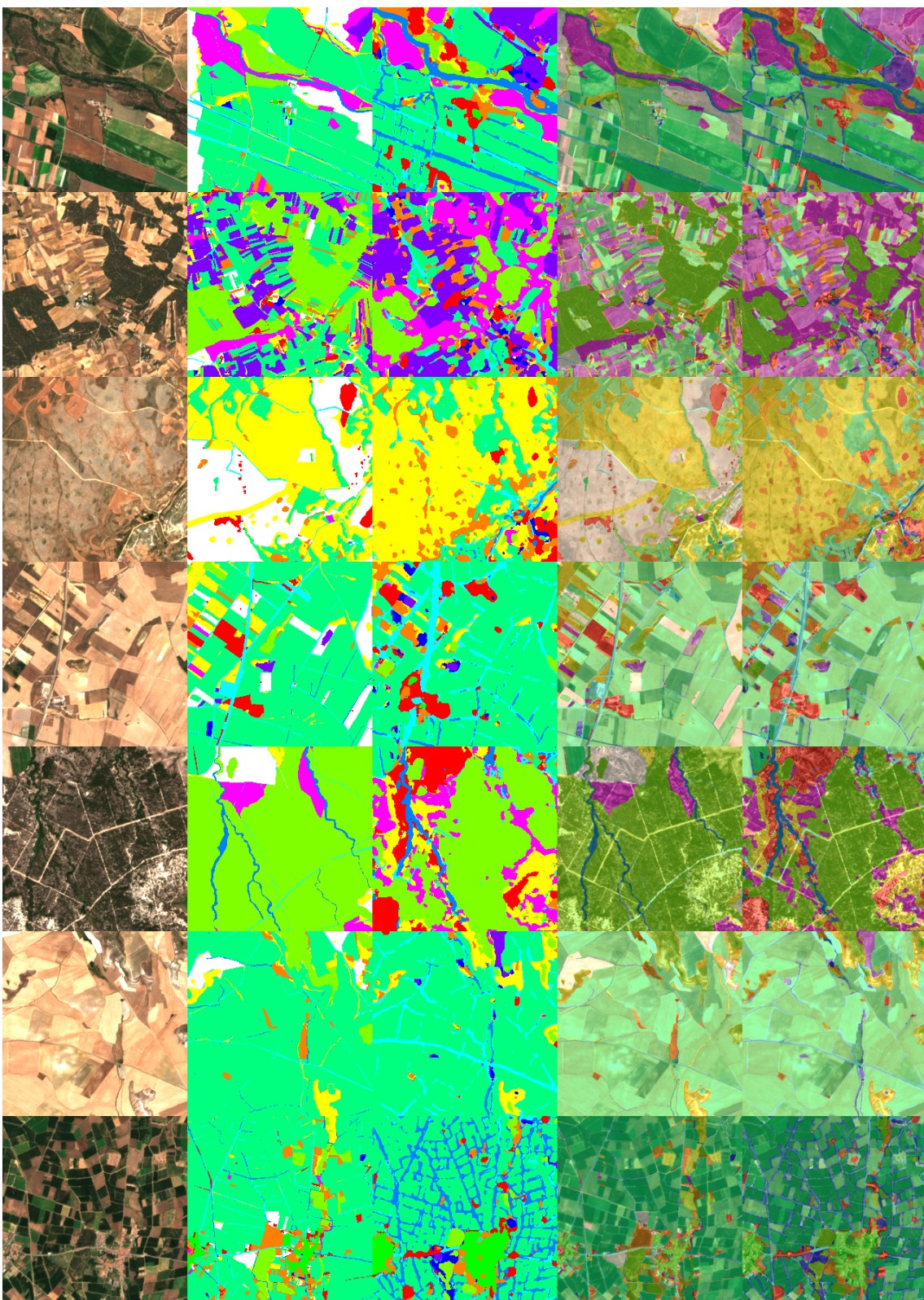

**Figure 14.** Visualization of the predicted results for UNet evaluated in UOS2 (Experiment Base). (**1st col.**) Original images, (**2nd col.**) ground truth masks, (**3rd col.**) predictions, (**4th col.**) overlapping of original images with ground truth masks, (**5th col.**) overlapping of original images with predictions.

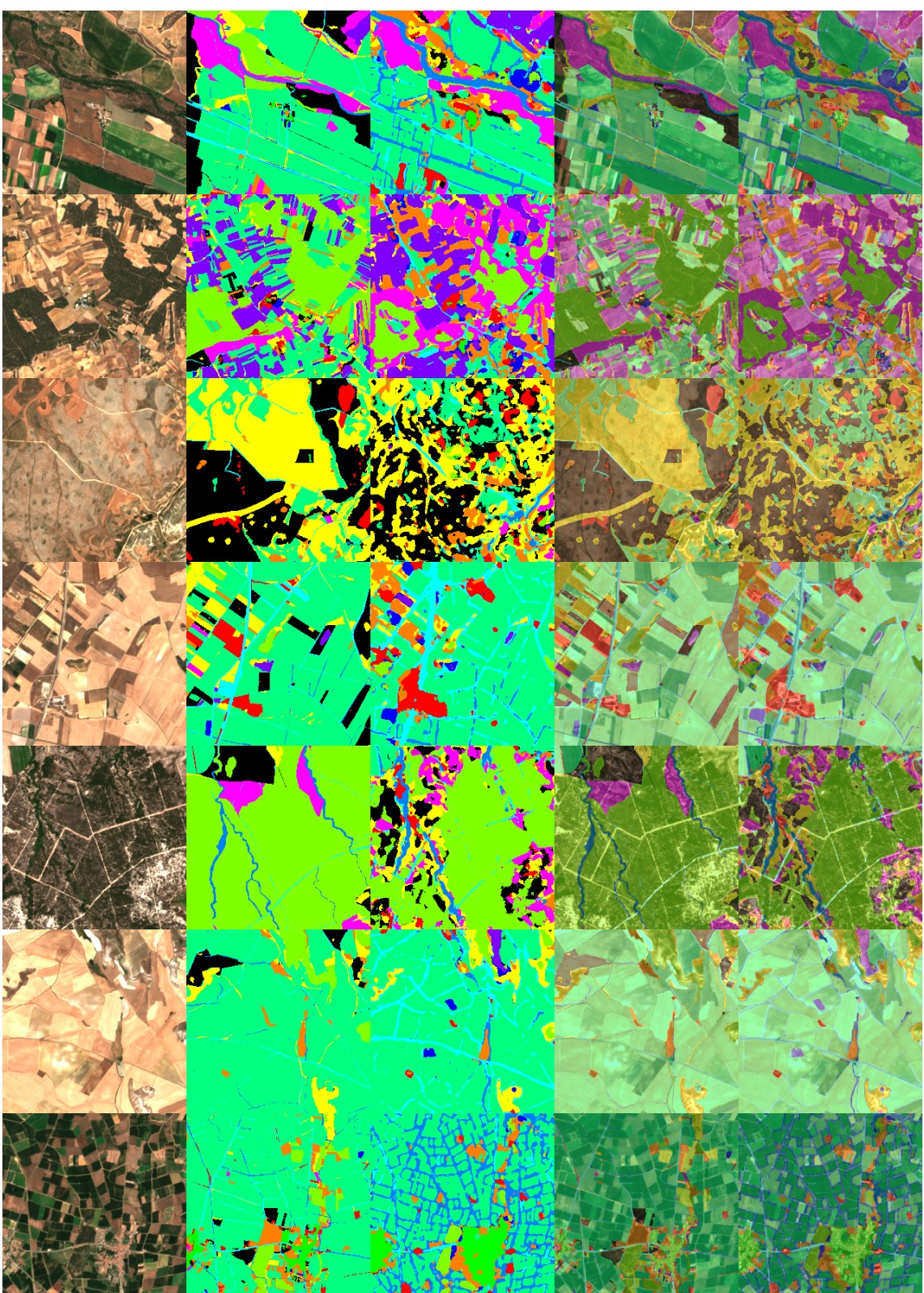

**Figure 15.** Visualization of the predicted results for UNet evaluated in UOS2 with "Other" class (Experiment All-Purpose). (**1st col.**) Original images, (**2nd col.**) ground truth masks, (**3rd col.**) predictions, (**4th col.**) overlapping of original images with ground truth masks, (**5th col.**) overlapping of original images with predictions.

After training the model, an evaluation with the test data was carried out. The results obtained can be seen in Table 20. "Base" and "All-Purpose" experiments (with and without class "Other") are carried out to compare with previous experiments.

Globally, the results coincide with previous experiments in that the "Base" experiment outperforms the "All-Purpose" experiment. Merging and reducing the number of classes drastically improves the accuracy of the predictions, obtaining an improvement of almost 22% and 25% on *PA* and *UA*, respectively, for the best experiments.

**Table 20.** Global metrics for the experiment with UNet with simplified classes on UOS2.

| Experiment | *OA* | *PA* | *UA* | *IoU* | *F$_1$* |
|---|---|---|---|---|---|
| Base | 0.822 | 0.786 | 0.677 | 0.576 | 0.727 |
| All-Purpose | 0.650 | 0.639 | 0.578 | 0.402 | 0.607 |

Looking at the results per class from Table 21, outstandingly high *PA* and *UA* can be seen for every class except BURO. As seen in previous experiments, when the "Other" class is used, all the remaining classes lose accuracy.

To prove that the tuning has been done correctly, progress from the loss function for training and validation can be seen in Figure 16. Validation loss stabilizes but training loss keeps lowering; if more epochs are executed, overfitting of the model would start to occur.

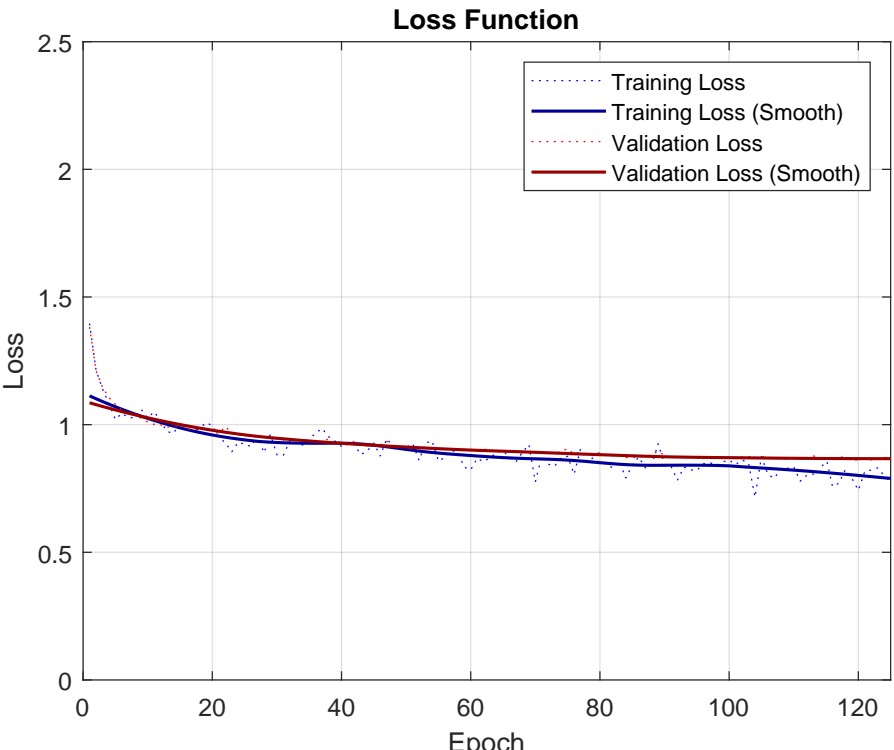

**Figure 16.** Loss function for the "Base" experiment with UNet for UOS2 with simplified classes.

**Table 21.** Class metrics for the experiment with UNet with simplified classes on UOS2. These classes consist of PASHGR (PA+SH+GR—all pastures), BURO (BU+RO—all infrastructures), ARVI (AR+VI—arable lands and vineyards), and OT.

| | Experiment Base | | | Experiment All-Purpose | | |
|---|---|---|---|---|---|---|
| Class | *PA* | *UA* | *IoU* | *PA* | *UA* | *IoU* |
| PASHGR | 0.82 | 0.81 | 0.69 | 0.71 | 0.52 | 0.43 |
| BURO | 0.70 | 0.25 | 0.23 | 0.70 | 0.17 | 0.16 |
| ARVI | 0.83 | 0.95 | 0.80 | 0.77 | 0.85 | 0.68 |
| OT | - | - | - | 0.36 | 0.75 | 0.21 |

To prevent confusion between the classes in Figures 17 and 18, a change in the color keys of classes has been made. In this visualization it is interesting to note that the class BURO, which performs the worst, has predictions around the boundaries of the plots. It seems to confuse the boundaries with roads. Furthermore, roads that are not classified as such in the original ground truth are predicted, outperforming the ground truth. This means that the low accuracy in this class can be seen as an error in the ground truth, which is logical given that not every dirt road is registered in the SIGPAC. This behavior may be beneficial to the predictions even though numerically the UA is low.

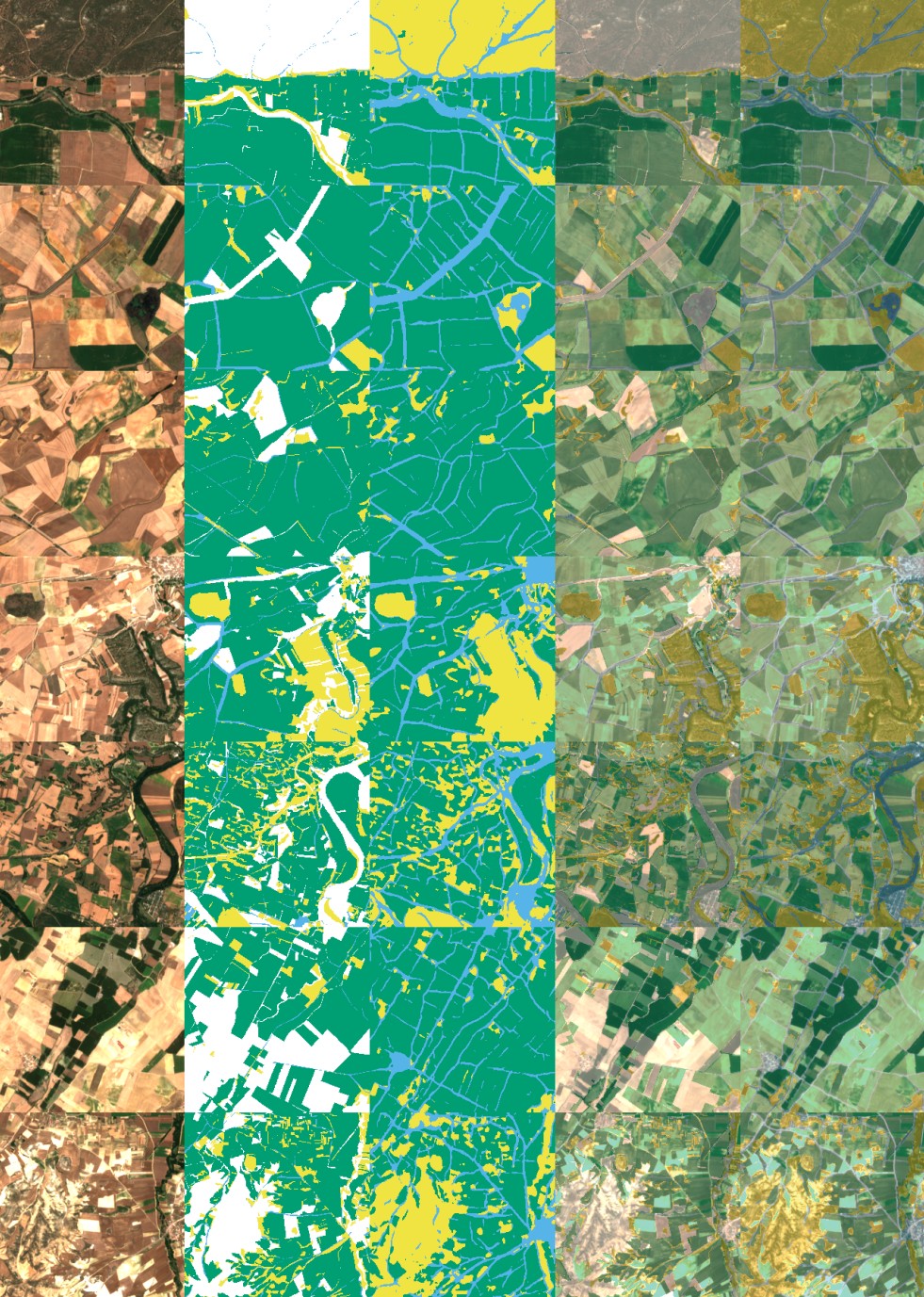

**Figure 17.** Visualization of the predicted results for UNet evaluated in UOS2 for four classes (Experiment Base). (**1st col.**) Original images, (**2nd col.**) ground truth masks, (**3rd col.**) predictions, (**4th col.**) overlapping of original images with ground truth masks, (**5th col.**) overlapping of original images with predictions.

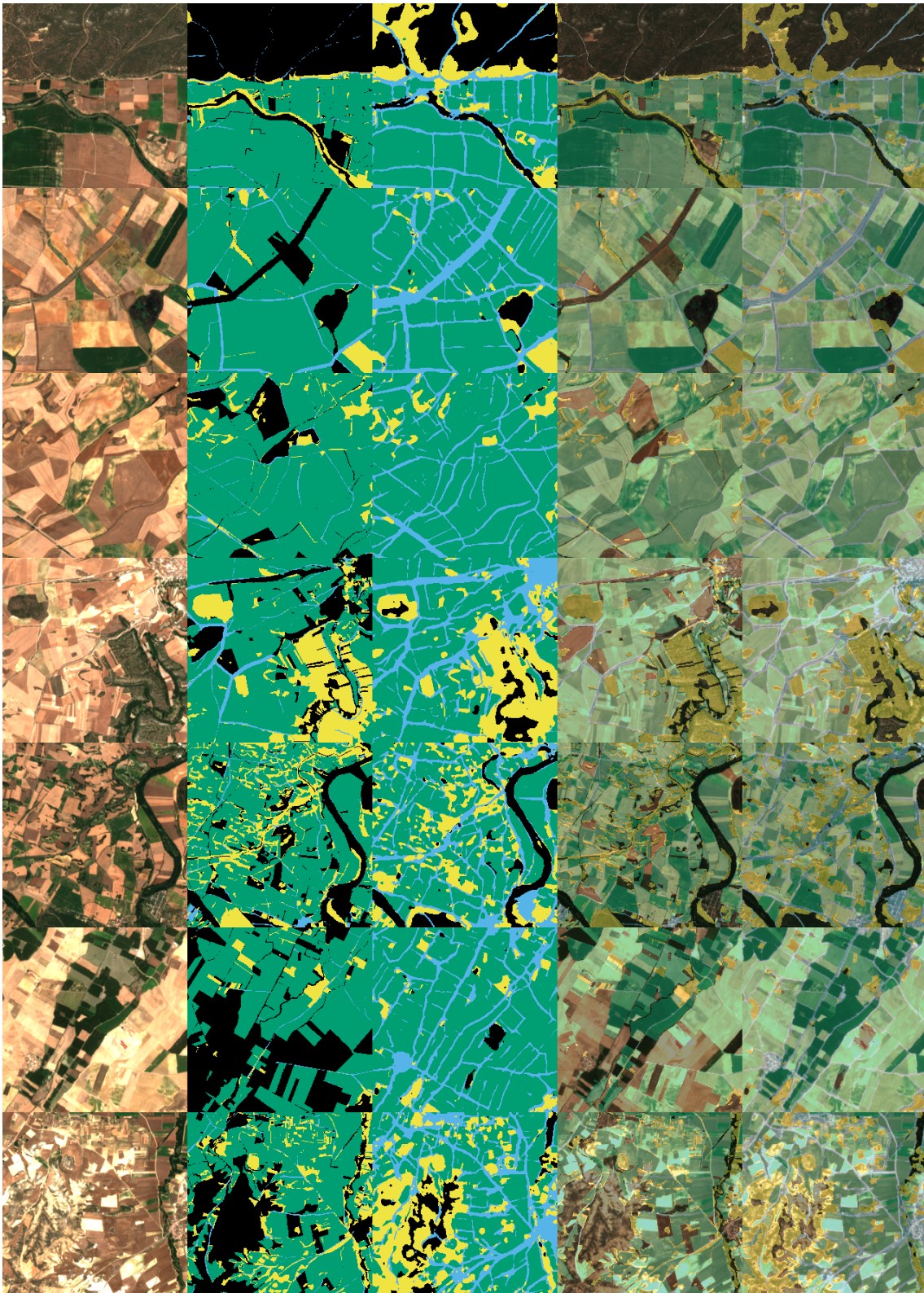

**Figure 18.** Visualization of the predicted results for UNet evaluated in UOS2 with four classes and "Other" class (Experiment All-Purpose). (**1st col.**) Original images, (**2nd col.**) ground truth masks, (**3rd col.**) predictions, (**4th col.**) overlapping of original images with ground truth masks, (**5th col.**) overlapping of original images with predictions.

*3.4. Deployment*

After training the models, a prototype for inference with a microservice architecture that follows modern standards using Flask was created. To ensure that all the required dependencies are met and to make deployment easier, Docker was used to containerize the microservice. Then, the different infrastructures were deployed to test performance. The model selected for this deployment is the UNet architecture with simplified classes from the "Base" experiment. The deep learning tool used for the inference is PyTorch.

Semantic segmentation networks can only operate with the same resolution used for training when the inference is done. This means that the only way to predict larger images is by cropping them and then fusing the predictions together. Each cropped image will take the same amount of time because every pixel in the image is predicted, so the same number of predictions are made for every image fed to the network. For this reason, the method used to test performance is based on the number of images requested per petition. These images have the same resolution as those used for training.

The majority of the tools used in Geographic Information Systems are designed to work with geojson as this format is easier to work with than a mask. To add realism to the service, a process to convert the output from the network to a georeferenced geojson with every region and land use type predicted on it, is added at the end of each petition. This adds many computations to the prototype as it must polygonize the mask from the output of the inference and convert it to a geojson. To limit the size the geojson, the Ramer–Douglas–Peucker algorithm [38] is executed to simplify the numbers of points defining the polygons. This process is adapted to make use of multiple CPU cores.

In Figure 19, performance by infrastructure is presented. "Local:A-B" are the machines used to train the networks in this work. These machines are connected via 1Gbit LAN. The rest of the infrastructures are provided by Microsoft Azure. Infrastructures "NC6", "A4 v2", "F4s v2", and "D4as v4" are IaaS. CaaS and FaaS implementations from Azure are also evaluated. In all the cases, the client is a local machine from outside Azure's network with a connectivity of approximately 250 Mbps.

For the evaluation of performance, times are defined as follows: "Load" is the time to access the libraries, loading the images into the memory, etc. "Prediction" is the time needed for the model to make the predictions. "Results" is the time required to convert the predictions into a geojson format. For this task, polygonization of the predictions masks, a simplification of their results using the algorithm RDP, and the generation of geojsons are timed. "Network" is the time needed to upload the images to be predicted and the time needed to download the predictions and their geojsons.

Performance for one image is presented in Figure 19. To compare performance between infrastructures, the latency ratio is shown on top of each stacked bar. This ratio is calculated as the total time of a given experiment divided by the total time of the best experiment. Cost-performance metrics for ten images are presented in Figure 20.

In Table 19, a minimal improvement when using GPU can be seen. "Load" times are negligible. "Prediction" depends on the single core speed and the number of cores. However, using a GPU is always faster. "Results" have the greatest impact on time, depending mainly on single core speed. "Network"is not affected by the infrastructure, except in the case of "Local:A-B" where both computers are in the same local network.

Table 20 shows almost no improvement when using a GPU, but the cost is multiplied. A tradeoff in cost and performance must be done since CaaS is more economical, but IaaS solutions such as "D4as v4" and "F4s v2" have better performance. "Local:A-B" seems to be the best option even when accounting for electricity cost and amortization over five years. Setting up and maintaining this kind of infrastructure can be excessively complex. However, cloud options get upgrades to the hardware from time to time. FaaS cost performance is calculated as if there were always a petition running for the entire hour. This means that the cost is zero if there are no petitions, but it would be greater if there were multiple simultaneous petitions causing multiple instances of FaaS to execute at the

same time, multiplying its cost. The rest of the infrastructures are priced for availability and not use: they can await petition at the same cost.

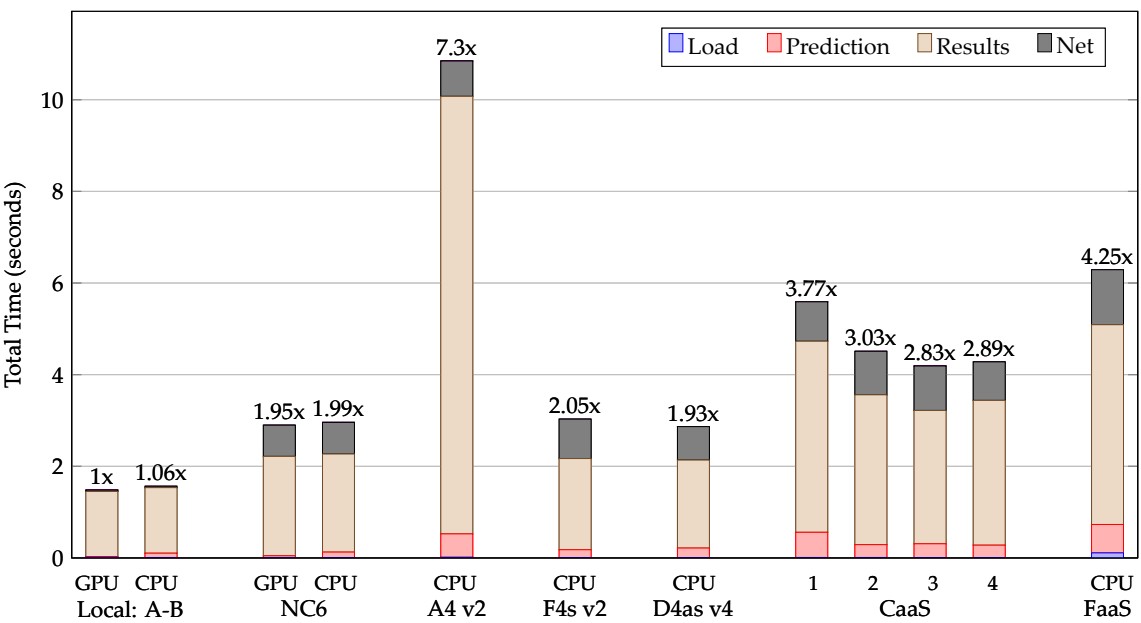

**Figure 19.** Performance in seconds per infrastructure.

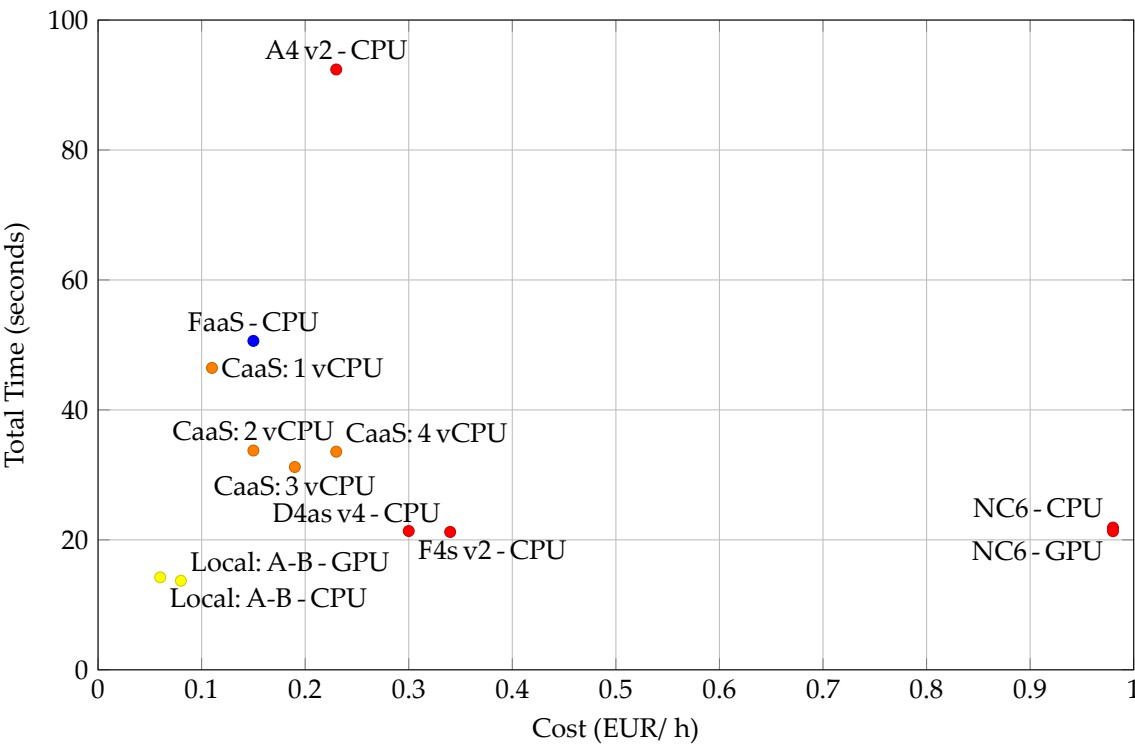

**Figure 20.** Cost performance per infrastructure. Prices and the computing power are subject to change (April 2021).

## 4. Conclusions

Land use classification in aerial imagery, especially satellite imagery, is a complex task. Thanks to recent advances in convolutional neural networks, a specific technology called semantic segmentation seems the most appropriate to improve on actual results. UNet is the most widely used semantic segmentation network thanks to its great flexibility, but when compared to DeepLabV3+, the latter performs far better. As a counterpoint,

DeepLabV3+ lacks an implementation that can use more than three bands, so it is the best option for RGB. However, for satellite imagery, where most of the data is presented as extra multi-spectral bands, it falls short. This gives the advantage to UNet for satellite imagery. Taking this into consideration, the most important factor for good accuracy in aerial datasets is their GSD. In the case of UOPNOA, impressive results were achieved with a GSD of 0.25 m/pixel (PA and UA of 78% and 75% respectively for DeepLabV3+ and 61% and 47% for UNet). However, in UOS2 using only RGB bands, its results are completely unusable with a PA and UA of less than 10%. UOS2 has a GSD of 10–60 m/pixel, which means that the difference in resolution is from 40 to 240 times worse.

When using UOS2 with at least six bands, including RGB and multi-spectral bands, its accuracy rises greatly (up to 48% PA and 37% UA), obtaining far better results but still falling short when compared to UOPNOA. However, this difference its greatly reduced when using more bands. While it is true that the GSD is of fundamental importance, the number and type of bands used are equally important. In this way, satellite imagery can equal the results of aircraft imagery. It is interesting to note that using more than six bands, even as many as thirteen, gives no significant improvement in the classification of land use.

Merging together similar or confusing classes improves the predictions noticeably (up to 78% PA and 67% UA for UNet with UOS2). This proves that it is better to use fewer, well defined classes. This can be verified with the experiment for UNet in UOS2 that uses only three classes, as this is one of the experiments with the best results even when compared with DeepLabV3+ on UOPNOA.

Using an all-purpose class is counterproductive, causing all the remaining classes to lose accuracy. There is a large variability associated with pixels that do not belong to any of the target classes. Semantic segmentation models will have to deal with these "unknown" classes when used in practice, unless the user removes these pixels beforehand. This could be done by selecting only the region of a desired plot, although this reduces the attractiveness of the approach. This approach is only useful if the input images have no pixels that belong to the target classes. Therefore, while this is acceptable for research purposes, only specific use cases can benefit from it.

The use of the two newly created datasets for land use classification in aerial imagery, UOPNOA and UOS2, are proven to be good for comparisons and evaluation of different models thanks to their great complexity and variability.

Finally, for performance with an inference prototype, network time alone takes more time than the predictions, even using only CPU. This is extremely important to take into account when choosing an infrastructure to offer services, like the one presented in this work. The use of a GPU is not recommended as it increases cost greatly, with no significant improvement in performance.

**Author Contributions:** Conceptualization, O.D.P., D.G.L., D.F.G., and R.U.; methodology, O.D.P.; software, O.D.P.; validation, D.G.L., D.F.G., and R.U.; formal analysis, O.D.P., D.F.G., and R.U.; investigation, O.D.P. and D.G.L.; resources, D.F.G. and R.U.; data curation, O.D.P.; writing—original draft preparation, O.D.P.; writing—review and editing, O.D.P., D.F.G., and R.U.; visualization, O.D.P.; supervision, Á.A.; project administration, D.F.G.; funding acquisition, D.F.G. and Á.A. All authors have read and agreed to the published version of the manuscript.

**Funding:** This research was funded by SERESCO S.A under the contract FUO-20-018 and also by the project RTI2018-094849-B-I00 of the Spanish National Plan for Research, Development and Innovation.

**Institutional Review Board Statement:** Not applicable.

**Informed Consent Statement:** Not applicable.

**Data Availability Statement:** UOPNOA and UOS2 datasets are publicly available in the platform Zenodo with the following DOI (last accessed on 10 June 2021): doi.org/10.5281/zenodo.4648002.

**Conflicts of Interest:** The authors declare no conflict of interest.

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
