# Peer review of "Evaluation of Semantic Segmentation Methods for Land Use with Spectral Imaging Using Sentinel-2 and PNOA Imagery"

_remotesensing, doi:10.3390/rs13122292_

Round 1

Reviewer 1 Report

The manuscript presents a study on the use of semantic segmentation for classification of agricultural areas. The study provides some interesting results, and the experiments were mostly well designed. However, there are a few aspects that are lacking and need to be properly addressed, as detailed below.
- English needs some polishing. There are a few oddly constructed sentences.
- Line 31: stating that deep learning based techniques are superior to other types of techniques (e.g. Random Forest) without proper qualification is not correct. CNN-based techniques can indeed outperform other types of techniques, especially when data is widely available and covers the entirety of the variability associated to a given problem, but may be lacking under more restrict data conditions.
- The structure of the introductory section is not adequate. The text should follow a logic sequence in order to guide the reader through the motivations and contributions of the work, but this is not the case. There are two main reasons for this. First, there are large portions of this section that belong in other parts of the text, such as lines 64-80 and 105-112, while other parts are overly long and should be condensed (e.g. lines 87-104). Second, the limitations of previous methods should be further explored in order to highlight the advantages of the techniques investigated, thus providing a better justification for the research.
- Line 204: no image has “perfect quality” in the real world. The authors should avoid this type of statement.
- Line 239: what do the authors mean by “unused”? It makes sense to group all pixels that do not belong to the classes of interest, but ignoring pixels is not possible when a model is used under real conditions, so keeping some pixels unused seem inadequate.
- Section 3.3: the datasets should be described in more detail. In particular, it would be useful to know how many pixels in total belong to each class. In addition, a detailed explanation of how the annotation of the images was carried out should be included, especially considering that the annotation process is prone to subjectivity, bias and other kinds of cognitive error (which may be partially compensated by using multiple annotators).
- Lines 353-357: ignoring pixels outside the regions of interest generates models that will not be prepared to deal with images in which those pixels were not manually removed, otherwise it will try to classify those pixels according to the classes for which it was trained. This severely limits the appeal of this approach, as the process is no longer automatic. It is clear that the dataset does not have enough images to properly cover the variability of the “other” class, which explains the relatively poor results, but again, simply ignoring those pixels in the training will not make the models any more useful in practice. The same observation is valid for the UNet and the Sentinel images.
- Table 17: the classes should be defined in the caption.
- The manuscript lacks a proper discussion section in which relevant aspects that do not become apparent by simply reading the results are treated with more detail. The problem of the “other” class seems particularly relevant. The variability associated to non-agricultural regions is certainly very large, and variations only tend to increase as other geographical regions are considered. While it is ok to ignore those regions for research purposes, semantic segmentation models will have to deal with these “undesirable” classes when used in practice, unless the user removes those pixels beforehand, which certainly reduces the appeal of the technique. This issue needs to be carefully addressed for proper characterization of the potential use of the models investigated by the authors.

Reviewer 2 Report

1) Title contradicts with content: the satellite and aerial images are applied in neural network training, therefore "spectral imaging" is more appropriate. The [227-229] categories are more land use classes neither crop, because under crop classification the agriculture species are waited neither roads and gardens. Deployment is related with software development neither science.

2) [3, 30-31, 87] "RF is mostly widely used": firstly, the conclusions are based on 2015 and 2000 sources. Secondly, the comprehensive background analysis of existing deep learning solutions must provided to make these summaries. There are plenty land-use modern solutions, which can be found in the same Remote Sensing journal: DeepUNet, ResUNet, etc. Additionally with background analysis, it is incorrect to make conclusion about state-of-art [81-86]

3) 2nd chapter provides basic/tutorial information about classification categories. The Q1 journal scientific articles are read by appropriate experts neither everyone.

4) [60-63] the aim of articles is to evaluate different sources of images. This task is totally uncompleted, because article describes service development for land use classification. (Additionally, it contradicts with title).

5) [14] "fast enough" is subjective parameter. It is related with Software Requirements Specification neither science.

6) [65-66] What does it means "up to 13 bands, including multi-spectral data"? Bands are spectral data.

7) [66-67] UOS2 is based on Sentinel-2. How images can be 10m? There are table 1, which depicts, that there are different resolutions for bands.

8) [68-69] Comparison of remote data sources is not complex. The specification tables are provided for images, which applied for comparison. The content is complex for analysis, but normally the appropriate region is selected for specific analysis tasks.

9) [67] What does it means "between the center"? Center is point.

10) 3.1.chapter. It can not be called "Analysis of different networks", because only two are provided, which can be compared. (Additionally - architectures neither networks).

11) [208] GoeTiff is not Tif extention, it is extended Tif with geospatial metadata.

12) [216-217] "widely used" - must be proved by literature analysis.

13) [Table 1] is Sentinel-2 specification, which can be found online, but PNOA image specification (which is only in Spain) is not provided. Additionally, it contains some mentions about NIR, that is strongly important.

14) [223-224] References into expert conclusions must be provided.

15) 4.1.1. ch. does not provide results with UOS2 dataset.

16) [320-324] Does DeepLabV3+ pretrained?

17) 3.5.ch. does not provides the structure diagram of trained models with layer parameters. Additionally, it does not provide information about deep learning tools.

18) Result tables [ex. t12-t13] does not provide all metric parameters described in 3.4.ch. Additionally, F1 is used for comparison for segmentation due to workbench datasets of competitions.

19) 4.3.ch. Deployment must provide 4 charts, because 2 neural network models was developed. Which one was tested, it is not provided.

20) [327-328]  The experiment plan proposes only UNet and DeepLabV3+, which are developed for segmentation. MobileNet and ResNet are classification architectures. Why they are depicted? Additionally, the experiments with these architectures must be proved by statistical results.

21) [Figures] "superposition" is called "overlaping" in GIS.

22) [285-266] the tuning must be proved by chart with acc/val_acc and/or loss/val_loss functions.

23) It is not sufficient to train existing neural network model for research. Article simply describes engineer solution, but it is not based on some comprehensive research to get new knowledge or to develop some new solution/ architecture.

Reviewer 3 Report

The manuscript applies and compares two different deep learning algorithms for crop classification.  Additionally, the authors published airborne and satellite datasets and their ground truth data along with this study.  I found the manuscript interesting; however, the authors present their study poorly. My major comments are as below:

  1. The manuscript requires extensive English editing
  2. Authors should choose a better title. The current title is very general.
  3. The introduction is very long and could be shortened. I also recommend authors merge sections 1 and 2 and avoid repeating the same content.
  4. Ln 39-40: Any reference?
  5. Ln 54-65: Any reference?
  6. Ln 148: Any reference or evidence to support this statement?
  7. Authors should go through their manuscript and add references for any bold statement which has not been investigated in their study.
  8. Material and methods need to be restructured. I am suggesting authors reorder their writing in this order: First, describe their Dataset with more details; also, they can merge the source of images with dataset description. Then, present their algorithms—finally, performance metrics.
  9. The authors should describe the information about their airborne data collection campaigns in the manuscript.
  10. The authors should provide a graphical overview of each architecture.
  11. Authors also should evaluate the performance of conventional machine learning algorithms like RF and SVM and compare their results with their proposed algorithms.
  12. Network parameters should be presented in material and methods.

Round 2

Reviewer 1 Report

My concerns were properly addressed.

Author Response

Thank you.

Reviewer 2 Report

  1. Traditionally point-to-point response letter is provided;
  2. Title contains point in the end (error);
  3. The provided diagrams of UNet and DeepLabV3+ can be found in the original papers. The request was to depict the architecture of trained CNN models;
  4. [69] "two of the best-known semantic segmentation architectures," - best-known is not correct feature to be selected for experiment. The related works were not reviewed, if it would be completed, you could to conclude about selected architecture;
  5. Will be cost-performance diagram true after some years, if price policy is changed? (the experiment time period must provided and cloud service in caption of figure). But it does not disclose the fact, what new knowledge for other scholars is provided? It is strongly individual economic choice for your solution;
  6. [326-328] mention about RF and SVM. These experiments were completed by authors? Why it was not mentioned in the experiment design description? Additionally, why time is provided? Have they been trained until end or it is temporary results in the training time?
  7. [323-324] "Both methods need much less data than a common neural network and take much longer to run." - RF is static-based method.
  8. Traditionally authors compare their obtained accuracy with other results to understand the accuracy sufficience for the-state-of-art.
  9. The UNet image is very similar to picture provided in the original paper - the reference is required or mentioning that it was drawn based on the example of source.

Reviewer 3 Report

The Authors addressed all my concerns. The manuscript in its current form can be considered for publication in remote sensing.

Author Response

Thank you.